# ENERGY-REGULARIZED SEQUENTIAL MODEL EDITING ON HYPERSPHERES

**Qingyuan Liu**[1]*, **Jia-Chen Gu**[2]*, **Yunzhi Yao**[3], **Hong Wang**[4], **Nanyun Peng**[2]
[1]Columbia University   [2]University of California, Los Angeles
[3]Zhejiang University   [4]University of Science and Technology of China
ql2505@columbia.edu, gujc@ucla.edu, violetpeng@cs.ucla.edu

🌐 Website   🔘 Code

## ABSTRACT

Large language models (LLMs) require constant updates to remain aligned with evolving real-world knowledge. Model editing offers a lightweight alternative to retraining, but sequential editing that updates the LLM knowledge through multiple successive edits often destabilizes representations and induces catastrophic forgetting. In this work, we seek to better understand and mitigate performance degradation caused by sequential editing. We hypothesize that *hyperspherical uniformity*, a property that maintains uniform distribution of neuron weights on a hypersphere, helps the model remain stable, retain prior knowledge, while still accommodate new updates. We use Hyperspherical Energy (HE) to quantify neuron uniformity during editing, and examine its correlation with editing performance. Empirical studies across widely used editing methods reveals a strong correlation between HE dynamics and editing performance, with editing failures consistently coinciding with high HE fluctuations. We further theoretically prove that HE dynamics impose a lower bound on the degradation of pretrained knowledge, highlighting why HE stability is crucial for knowledge retention. Motivated by these insights, we propose SPHERE (**S**parse **P**rojection for **H**yperspherical **E**nergy-**R**egularized **E**diting), an HE-driven regularization strategy that stabilizes neuron weight distributions, ultimately preserving prior knowledge while enabling reliable sequential updates. Specifically, SPHERE identifies a sparse space complementary to the principal hyperspherical directions of the pretrained weight matrices and projects new knowledge onto it, attenuating perturbations on the principal directions. Extensive experiments on LLaMA3 (8B) and Qwen2.5 (7B) show that SPHERE outperforms the best baseline in editing capability by an average of 16.41%, while most faithfully preserving general model performance, thereby offering a principled path toward reliable large-scale knowledge editing.

## 1 INTRODUCTION

Large language models (LLMs) have demonstrated strong capabilities in knowledge storage, reasoning, and generation (DeepSeek-AI et al., 2024; Meta AI, 2024; Yang et al., 2025; OpenAI, 2025). However, the knowledge embedded in LLMs inevitably becomes outdated or incorrect, as real-world facts continuously evolve (Ji et al., 2023; Huang et al., 2025). Retraining LLMs to incorporate such updates is prohibitively expensive, motivating the development of *model editing* (also known as *knowledge editing*) (Cao et al., 2021; Mitchell et al., 2022; Meng et al., 2023; Gu et al., 2024; Fang et al., 2025). The most practical setting for model editing is *sequential editing*, where multiple updates are applied over time. However, previous studies have shown that such interventions often suffer from significant performance degradation due to catastrophic forgetting (Gu et al., 2024; Gupta et al., 2024). Consequently, reconciling the trade-off between preserving original pretrained knowledge and integrating new editing knowledge remains an unresolved challenge.

In this work, we seek to better understand and mitigate the performance degradation caused by sequential editing. We revisit model editing from the perspective of *hyperspherical uniformity* of perturbed weights (Liu et al., 2021), motivated by the observation that sequential edits often disrupt

---

* Equal contribution.

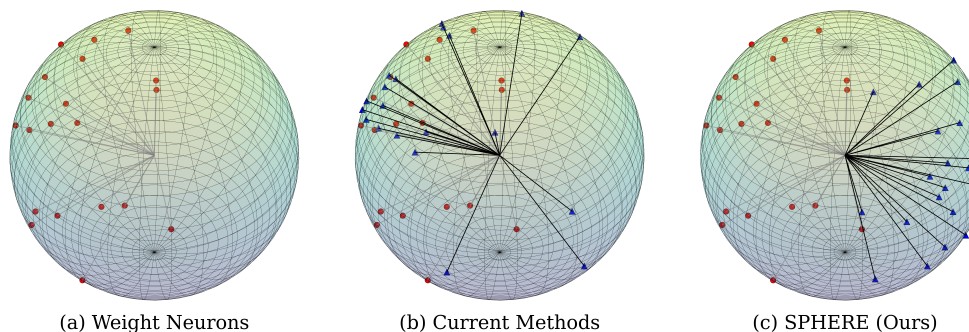

(a) Weight Neurons    (b) Current Methods    (c) SPHERE (Ours)

Figure 1: (a) A weight matrix is viewed as a set of neurons (red dots) on a hypersphere. (b) Current SOTA methods (Ma et al., 2025; Fang et al., 2025) introduce perturbations (blue triangles) that interfere with the principle hyperspherical directions of pre-edit weights. (c) SPHERE projects new knowledge onto a sparse space complementary to the principal hyperspherical directions.

weight geometry, leading to degraded representations. Previous studies have shown that viewing a weight matrix as a set of neurons on a hypersphere (as shown in Figure 1 (a)) and maintaining their hyperspherical uniformity is crucial for stable training and effective generalization (Cogswell et al., 2016; Xie et al., 2017a; Qiu et al., 2023). To investigate the applicability of these principles to sequential editing, we adopt *hyperspherical energy (HE)* (Liu et al., 2018; Qiu et al., 2023) as a measure to quantify weight uniformity throughout sequential editing. HE calculates the dispersion of neuron weight vectors on a hypersphere, where lower energy corresponds to a more balanced distribution of neurons. By tracking HE dynamics throughout sequential editing, we can better understand how edits affect weight uniformity, identify early signs of destabilization, and even develop HE-driven regularization strategies to stabilize the editing process.

To reveal the mechanisms underlying successful editing strategies from the perspective of hyperspherical uniformity, we first empirically analyze how HE evolves throughout sequential editing and examine how these dynamics relate to editing performance across six widely used methods. Experimental results reveal a strong correlation between hyperspherical uniformity and editing performance, with editing failures consistently coinciding with its collapse. Meanwhile, more advanced editing methods have proven more effective at preserving hyperspherical uniformity. To complement these empirical findings, we further provide a theoretical analysis verifying that variations in HE establish a lower bound on the interference with the original pretrained knowledge. This result clarifies that state-of-the-art (SOTA) editing methods implicitly regulate hyperspherical uniformity and the lower bound on the interference, providing a principled explanation for their enhanced robustness.

Motivated by these empirical and theoretical findings, we propose SPHERE (**S**parse **P**rojection for **H**yperspherical **E**nergy-**R**egularized **E**diting), an HE-driven regularization strategy that stabilizes neuron weight distributions, ultimately preserving prior knowledge while enabling reliable sequential updates. The key insight is that, as shown in Figure 1 (b), current methods often introduce perturbations that interfere with the principal hyperspherical directions of the pretrained weight matrices, leading to instability, loss of uniformity, and eventual degradation of model performance. To counteract these side effects, as shown in Figure 1 (c), SPHERE identifies a sparse space complementary to the principal hyperspherical directions of the pretrained weight matrices and projects new knowledge onto it, attenuating perturbation components aligned with those principal directions. By doing so, SPHERE effectively preserves the hyperspherical uniformity and substantially extends the number of effective sequential edits.

We evaluated SPHERE on **two LLMs**, including LLaMA3 (8B) (AI@Meta, 2024) and Qwen2.5 (7B) (Team, 2024), on **two editing datasets**, including CounterFact (Meng et al., 2022) and ZsRE (Levy et al., 2017). **Four downstream tasks** including reasoning (Cobbe et al., 2021), natural language inference (Dagan et al., 2005), open-domain QA (Kwiatkowski et al., 2019), and closed-domain QA (Clark et al., 2019) are employed to demonstrate the impact of editing on the general abilities of LLMs. Experimental results show that SPHERE sustains editing capacity under large-scale editing settings, outperforming the best baseline (Fang et al., 2025) by **16.41%** on average. Beyond editing capacity, it more effectively preserves the hyperspherical uniformity and the general abilities of edited models than all baselines. Furthermore, as a plug-and-play enhancement, SPHERE improves the mainstream editing methods (Meng et al., 2023; Gu et al., 2024; Ma et al., 2025) by **38.71%** on average, offering a principled path toward reliable and scalable editing.

## 2 PRELIMINARIES

### 2.1 MODEL EDITING

Sequential model editing aims to update the knowledge stored in LLMs through multiple successive edits. Each edit modifies the model parameter $\boldsymbol{W} \in \mathbb{R}^{d_1 \times d_0}$ by adding a perturbation $\Delta \in \mathbb{R}^{d_1 \times d_0}$ in a locate-then-edit paradigm (Meng et al., 2022), where $d_0$ and $d_1$ represent the dimensions of the intermediate and output layers of the feed-forward network (FFN), respectively. Specifically, suppose each edit updates $u$ pieces of knowledge in the form of (subject $s$, relation $r$, object $o$), e.g., ($s$ = *United States*, $r$ = *President of*, $o$ = *Donald Trump*). The perturbed parameter is expected to associate $u$ new *key-value* ($k$-$v$) pairs, where $k$ and $v$ encode $(s, r)$ and $(o)$ of the new knowledge, respectively. We can stack these keys and values into matrices as follows:

$$\boldsymbol{K}_1 = [\boldsymbol{k}_1 \,|\, \boldsymbol{k}_2 \,|\, \ldots \,|\, \boldsymbol{k}_u] \in \mathbb{R}^{d_0 \times u}, \quad \boldsymbol{V}_1 = [\boldsymbol{v}_1 \,|\, \boldsymbol{v}_2 \,|\, \ldots \,|\, \boldsymbol{v}_u] \in \mathbb{R}^{d_1 \times u}, \tag{1}$$

where the subscripts of $\boldsymbol{k}$ and $\boldsymbol{v}$ represent the index of the to-be-updated knowledge. Therefore, the editing objective can be expressed as:

$$\Delta \boldsymbol{W} = \arg \min_{\Delta \hat{\boldsymbol{W}}} \left\| (\boldsymbol{W} + \Delta \hat{\boldsymbol{W}}) \boldsymbol{K}_1 - \boldsymbol{V}_1 \right\|^2, \tag{2}$$

where $\| \cdot \|^2$ denotes the sum of the squared elements in the matrix.

Additionally, current methods typically incorporate an error term to preserve the original knowledge. Let $\boldsymbol{K}_0$ and $\boldsymbol{V}_0$ represent the matrices formed by stacking the $\boldsymbol{k}$ and $\boldsymbol{v}$ corresponding to the original pretrained knowledge. Eqn. 2 is regularized by involving the error term as follows:

$$\Delta \boldsymbol{W} = \arg \min_{\Delta \hat{\boldsymbol{W}}} \left( \left\| (\boldsymbol{W} + \Delta \hat{\boldsymbol{W}}) \boldsymbol{K}_1 - \boldsymbol{V}_1 \right\|^2 + \left\| (\boldsymbol{W} + \Delta \hat{\boldsymbol{W}}) \boldsymbol{K}_0 - \boldsymbol{V}_0 \right\|^2 \right). \tag{3}$$

Since $\boldsymbol{K}_0$ and $\boldsymbol{V}_0$ encode the original pretrained knowledge, we have $\boldsymbol{W} \boldsymbol{K}_0 = \boldsymbol{V}_0$ (cf. Eqn. 1). By applying the normal equation, if the closed-form solution of Eqn. 3 exists, it can be written as:

$$\Delta \boldsymbol{W} = (\boldsymbol{V}_1 - \boldsymbol{W} \boldsymbol{K}_1) \boldsymbol{K}_T^\top \left( \boldsymbol{K}_0 \boldsymbol{K}_0^\top + \boldsymbol{K}_1 \boldsymbol{K}_1^\top \right)^{-1}. \tag{4}$$

Since the full scope of an LLM's knowledge is generally inaccessible, $\boldsymbol{K}_0$ is difficult to obtain directly but can be approximated from abundant text input. See Appendix B for more details.

### 2.2 HYPERSPHERICAL ENERGY

Hyperspherical Energy (HE) serves as a quantitative metric for measuring *hyperspherical uniformity*. Given a group of neurons, HE characterizes their uniformity on a hypersphere by defining a generic potential energy based on their pairwise relationship. Lower energy represents that these neurons are more diverse and uniformly distributed, while higher energy reflects redundancy. Given a weight matrix $\boldsymbol{W} \in \mathbb{R}^{N \times (d+1)}$ represented as a set of $N$ neurons (i.e., kernels), where each row $\boldsymbol{w}_i \in \mathbb{R}^{d+1}$ corresponds to a neuron, its HE is defined as:

$$\boldsymbol{E}_{\boldsymbol{s}, \boldsymbol{d}} \left( \hat{\boldsymbol{w}}_i \,|_{i=1}^N \right) = \sum_{i=1}^N \sum_{j=1, j \neq i}^N f_s \left( \| \hat{\boldsymbol{w}}_i - \hat{\boldsymbol{w}}_j \| \right) = \begin{cases} \sum_{i \neq j} \| \hat{\boldsymbol{w}}_i - \hat{\boldsymbol{w}}_j \|^{-s}, & s > 0 \\ \sum_{i \neq j} \log \left( \| \hat{\boldsymbol{w}}_i - \hat{\boldsymbol{w}}_j \|^{-1} \right), & s = 0 \end{cases} \tag{5}$$

where $\| \cdot \|$ denotes Euclidean distance, $f_s(\cdot)$ is a decreasing real-valued function, and $\hat{\boldsymbol{w}}_i = \frac{\boldsymbol{w}_i}{\| \boldsymbol{w}_i \|}$ is the $i$-th neuron weight projected onto the unit hypersphere $\mathbb{S}^d = \{ \boldsymbol{w} \in \mathbb{R}^{d+1} \mid \| \boldsymbol{w} \| = 1 \}$. We also denote $\hat{\boldsymbol{W}}_N = \{ \hat{\boldsymbol{w}}_1, \cdots, \hat{\boldsymbol{w}}_N \in \mathbb{S}^d \}$, and $E_s = \boldsymbol{E}_{\boldsymbol{s}, \boldsymbol{d}}(\hat{\boldsymbol{w}}_i \,|_{i=1}^N)$ for short. There are plenty of choices for $f_s(\cdot)$, but in this paper we use $f_s(z) = z^{-s}$, $s > 0$, known as Riesz $s$-kernels. Since each $\hat{\boldsymbol{w}}_i$ lies on the unit hypersphere, the squared Euclidean distance between two neurons can be equivalently expressed in angular form as $\| \hat{\boldsymbol{w}}_i - \hat{\boldsymbol{w}}_j \|^2 = 2(1 - \cos \theta_{ij})$, where $\theta_{ij}$ is the angle between $\hat{\boldsymbol{w}}_i$ and $\hat{\boldsymbol{w}}_j$. Substituting this into Eqn. 5, we have:

$$\boldsymbol{E}_{\boldsymbol{s}, \boldsymbol{d}} \left( \hat{\boldsymbol{w}}_i \,|_{i=1}^N \right) = \sum_{i=1}^N \sum_{j=1, j \neq i}^N \left( 2(1 - \cos \theta_{ij}) \right)^{-s/2}. \tag{6}$$

This angular formulation highlights the geometric interpretation of HE: a higher value corresponds to neuron clustering with low angular diversity, while a lower value reflects a more uniform angular distribution across the hypersphere.

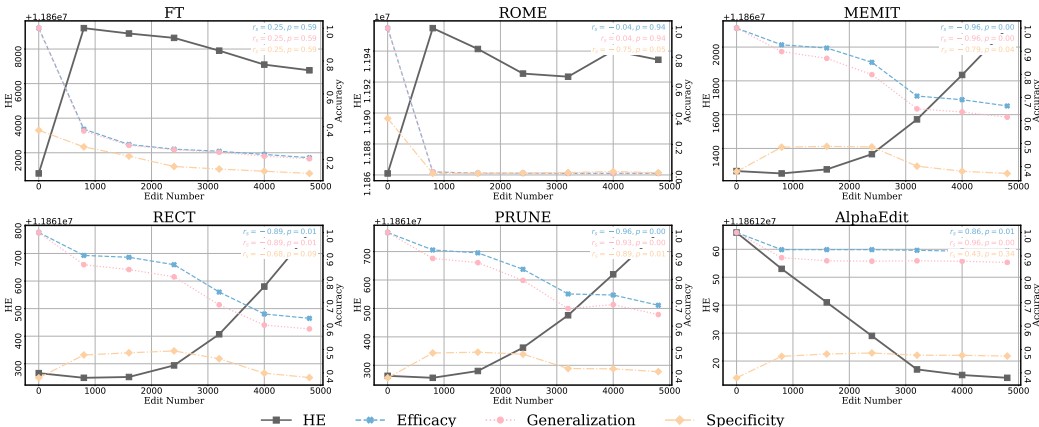

Figure 2: Trends of HE and editing performance throughout sequential editing. The Spearman correlation scores between HE and each editing metric displayed at the end of each curve.

# 3 CORRELATION BETWEEN HYPERSPHERICAL UNIFORMITY AND EDITING

HE and model editing are intrinsically connected through their shared focus on the geometry of high-dimensional parameter spaces. An optimal HE corresponds to more uniformly distributed representations on the unit hypersphere, typically reflecting well-conditioned parameters that enable reliable and stable sequential editing. We first present empirical evidence revealing a strong correlation between HE and editing stability (Section 3.1), followed by a formal theoretical analysis establishing the mathematical link between the two (Section 3.2).

## 3.1 EMPIRICAL ANALYSIS OF THE HE–EDITING STABILITY CORRELATION

To understand the failure modes of large-scale sequential editing, we examined how HE evolves throughout the editing process. We performed 5,000 sequential edits on ZsRE dataset (Levy et al., 2017) with a batch size of 100 on LLaMA3-8B (AI@Meta, 2024) using six widely used editing methods, including Fine-Tuning (FT) (Zhu et al., 2020), ROME (Meng et al., 2022), MEMIT (Meng et al., 2023), RECT (Gu et al., 2024), PRUNE (Ma et al., 2025), and AlphaEdit (Fang et al., 2025). After each edit, we computed the HE of the perturbed weights and evaluated the editing performance using well-established metrics, including **Efficacy** (edit success), **Generalization** (paraphrase success), and **Specificity** (neighborhood success). Readers can refer to Appendix D.2 for detailed definition of these metrics. We summarize our main observations as follows:

**Observation 1: Collapse in sequential editing is closely tied to sharp fluctuations in HE.** Figure 2 reveals a strong correlation between HE dynamics and editing performance. The Spearman correlation scores (Spearman, 1904) between HE and each editing metric, displayed at the end of each curve, consistently indicate a strong statistical dependence before model collapse[1]. Most methods collapse well before 3,000 edits, whereas AlphaEdit demonstrates the strongest long-term editing capacity with the best preservation of hyperspherical uniformity. A closer examination of the metrics shows a consistent pattern in which each drop in performance is consistently accompanied by rapid shifts in HE, underscoring its central role in maintaining sequential editing stability.

**Observation 2: Advanced editing methods suppress HE fluctuations effectively.** Figure 3 illustrates the correlation between changes in HE ($\Delta$HE) and editing performance ($\Delta$Acc.), where each point denotes the difference between two consecutive batch edits: points near the origin indicate greater stability with minimal variation in both HE and accuracy, while points farther away reflect larger fluctuations and less stable editing. Most advanced methods exhibit tightly clustered distributions near the origin, indicating stable editing dynamics and minimal weight distortion. Furthermore, we fit a linear regression over all points across metrics, which demonstrates a statistically

---

[1]Since FT and ROME rapidly collapse at the very beginning, we instead emphasize their correlations by examining the curve fluctuations.

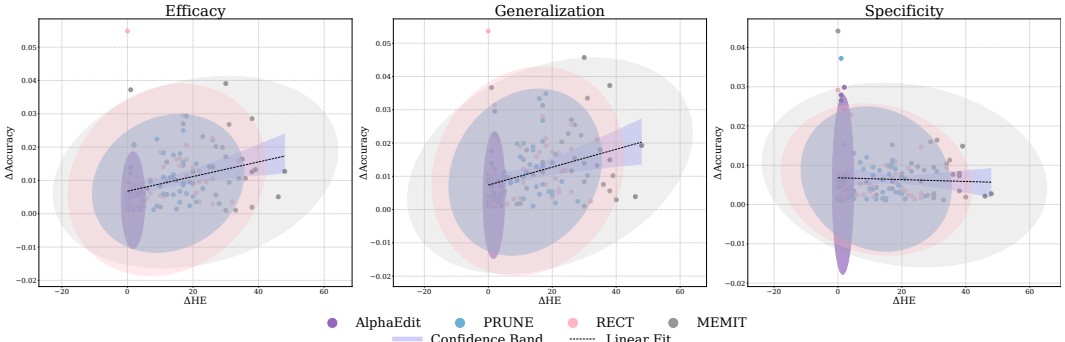

Figure 3: Correlation between changes in HE and editing performance across consecutive edited weights. Each point corresponds to a $\Delta$HE–$\Delta$Acc. pair for one method over five thousand sequential edits. Confidence ellipses and regression lines illustrate overall trends.

significant positive correlation between $\Delta$HE and $\Delta$Acc. in terms of Efficacy and Generalization. This suggests a strong positive correlation between editing stability and HE stability, implying that the effectiveness of SOTA approaches may stem from their ability to suppress HE fluctuations.

## 3.2 THEORETICAL ANALYSIS OF HE'S IMPACT ON EDITING STABILITY

We further turn to a theoretical analysis of how HE impacts editing stability, aiming to provide a principled explanation for the patterns observed in practice. Given the editing objective in Eqn. 2, it inevitably perturbs the original pretrained knowledge in LLMs, which can be expressed as:

$$\Delta \boldsymbol{V} = (\boldsymbol{W} + \Delta \boldsymbol{W})\boldsymbol{K}_0 - \boldsymbol{V}_0 = \Delta \boldsymbol{W}\boldsymbol{K}_0, \tag{7}$$

where $\boldsymbol{W}\boldsymbol{K}_0 = \boldsymbol{V}_0$, as $\boldsymbol{K}_0$ and $\boldsymbol{V}_0$ represent the original knowledge. Additionally, from the HE definition in Eqn. 5, the change in HE after editing can be written as:

$$\Delta \boldsymbol{H}\boldsymbol{E} = \sum_{i \neq j} \left( \|\boldsymbol{w}_i - \boldsymbol{w}_j\|^{-2} - \|\boldsymbol{w}_i + \Delta \boldsymbol{w}_i - \boldsymbol{w}_j - \Delta \boldsymbol{w}_j\|^{-2} \right), \tag{8}$$

where $\Delta \boldsymbol{w}_i$ denotes the perturbation to $\boldsymbol{w}_i$. This term $\Delta \boldsymbol{H}\boldsymbol{E}$ measures how angular separation among weight vectors changes after editing.

Our theoretical analysis, detailed in Appendix C.1, culminates in a key result that formally links the geometric change in weight space $\Delta \boldsymbol{H}\boldsymbol{E}$ to the output perturbation $\Delta \boldsymbol{V}$, derived from Proposition 2.

**Theorem 1** (Lower Bound on Output Perturbation). *Under the assumptions of orthonormal inputs and small perturbations, the output perturbation $\Delta \boldsymbol{V}$ is lower-bounded by squared change in HE:*

$$|\Delta \boldsymbol{V}| \geq \left( \frac{\Delta \boldsymbol{H}\boldsymbol{E}}{K} \right)^2, \quad K = 4 \left( \sum_{k=1}^{p} \left( \sum_{j \neq k} \|\boldsymbol{w}_k - \boldsymbol{w}_j\|^{-3} \right)^2 \right)^{1/2}. \tag{9}$$

*where $K$ is a constant dependent on the original weight matrix geometry.*

This theorem reveals a key insight: the change in HE ($|\Delta \boldsymbol{H}\boldsymbol{E}|$) inevitably induces a substantial output perturbation ($\Delta \boldsymbol{V}$), meaning that edits that significantly distort the geometric arrangement of neurons are bound to corrupt pretrained knowledge. This result provides a solid theoretical foundation for our empirical findings and underscores HE as a fundamental indicator of editing stability.

## 4 SPHERE

On account of the above findings, we argue that ideal sequential editing should preserve the hyperspherical uniformity of edited weights. Accordingly, we introduce SPHERE, an HE-driven regularization strategy designed to mitigate HE fluctuations while integrating new knowledge.

SPHERE first estimates the principal hyperspherical directions of pretrained knowledge and then defines their orthogonal complement as the sparse space. Projecting editing perturbations onto this space enables knowledge injection while minimizing interference with original knowledge.

**Principal Space Estimation** To identify the principal hyperspherical directions in $\boldsymbol{W}$, we seek a unit vector $v \in \mathbb{R}^d$ that maximizes the variance of all neurons in $\boldsymbol{W}$ when projected onto $v$ as:

$$\boldsymbol{v} = \arg \max_{\|\hat{\boldsymbol{v}}\|=1} \left( \tfrac{1}{n} \|\boldsymbol{W}\hat{\boldsymbol{v}}\|^2 \right) = \arg \max_{\|\hat{\boldsymbol{v}}\|=1} \left( \tfrac{1}{n} \hat{\boldsymbol{v}}^\top (\boldsymbol{W}^\top \boldsymbol{W}) \hat{\boldsymbol{v}} \right). \tag{10}$$

According to the Rayleigh quotient theory (Horn & Johnson, 1985; Parlett, 1998), the maximum of $\frac{1}{n} v^\top \boldsymbol{W}^\top \boldsymbol{W} v$ corresponds to the largest eigenvalue $\lambda^*$ of $\frac{1}{n} \boldsymbol{W}^\top \boldsymbol{W}$, with the associated eigenvector $v^*$ as the principal direction. Extending this to the top-$r$ principal directions enables us to capture a richer low-dimensional space of the weight geometry, so we collect the eigenvectors associated with the $r$ largest eigenvalues to form the principal space matrix, which can be expressed as:

$$\boldsymbol{U} = [\boldsymbol{v}_{d-r+1}, \ldots, \boldsymbol{v}_d] \in \mathbb{R}^{d \times r}, \tag{11}$$

where $r$ satisfies $\sum_{i=d-r+1}^{d} \lambda_i \geq \eta \sum_{i=1}^{d} \lambda_i$, with the cumulative ratio $\eta$ (see Appendix D.4.1).

**Sparse Space Definition** This space is defined as the orthogonal complement of $\boldsymbol{U}$ in Eqn. 11 as:

$$\boldsymbol{P}_\perp = I - \alpha \boldsymbol{U}\boldsymbol{U}^\top \in \mathbb{R}^{d \times d}, \tag{12}$$

where $\alpha$ controls the suppression strength of the components along the subspace spanned by $\boldsymbol{U}$ (see Appendix D.4.1). Specifically, $\alpha = 1$ corresponds to a hard orthogonal projection that completely removes the contribution of $\boldsymbol{U}$, while $0 < \alpha < 1$ yields a soft projection that only attenuates it.

**Sparse Space Projection** Given a perturbation matrix $\Delta \boldsymbol{W}$ produced by any editing method, we project it onto the sparse space using $P_\perp$, and then combine it with the original weight matrix as:

$$\hat{\boldsymbol{W}} = \boldsymbol{W} + \Delta \boldsymbol{W}_{\text{proj}} = \boldsymbol{W} + \Delta \boldsymbol{W} \boldsymbol{P}_\perp. \tag{13}$$

In summary, SPHERE suppresses perturbations aligned with the principal weight directions to preserve hyperspherical uniformity, enabling more stable, longer-lasting performance without compromising general abilities. For theoretical completeness, we also provide a mathematical proof that SPHERE suppresses the $\Delta \boldsymbol{H}\boldsymbol{E}$, ensuring bounded variations in the hidden representations $\Delta \boldsymbol{V}$ and justifying its effectiveness during editing (see Appendix C.2). More details in Appendix D.5

## 5 EXPERIMENTS

In this section, we aim to address the following research questions:

- **RQ1:** How does SPHERE perform on sequential editing tasks compared to baseline methods?
- **RQ2:** Can SPHERE effectively preserve the hyperspherical uniformity of edited weights?
- **RQ3:** How does SPHERE-edited LLMs perform on general ability evaluations?
- **RQ4:** Can baseline methods be significantly improved with plug-and-play SPHERE?
- **RQ5:** What is the computational overhead of SPHERE in terms of time efficiency?
- **RQ6:** How do different hyperparameter choices affect the stability of SPHERE?

### 5.1 EXPERIMENTAL SETUP

**Base LLMs and Baseline Methods.** Experiments were conducted on LLaMA3 (8B) (AI@Meta, 2024) and Qwen2.5 (7B) (Team, 2024). We compared our approach against a range of representative sequential editing baselines, including Fine-Tuning (FT) (Zhu et al., 2020), MEMIT (Meng et al., 2023), RECT (Gu et al., 2024), PRUNE (Ma et al., 2025), and AlphaEdit (Fang et al., 2025).

**Datasets and Evaluation Metrics.** Two widely used benchmarks were adopted: Counter-Fact (Meng et al., 2022) and ZsRE (Levy et al., 2017). Following prior work (Meng et al., 2022), five evaluation metrics were reported: **Efficacy** (edit success), **Generalization** (paraphrase success), **Specificity** (neighborhood success), **Fluency** (generation entropy), and **Consistency** (reference score). For rigorous evaluation, we adopt the **average top-1 accuracy** as the metric for both datasets. Readers can refer to Appendix D for more detailed experimental setup.

Table 1: Comparison of SPHERE with existing methods on sequential editing. *Eff.*, *Gen.*, *Spe.*, *Flu.* and *Consis.* denote Efficacy, Generalization, Specificity, Fluency and Consistency, respectively. The best results are highlighted in bold, while the second-best results are underlined.

| Method | Model | ZSRE | | | Counterfact | | | | |
|--------|-------|------|------|------|------|------|------|------|------|
| | | Eff.↑ | Gen.↑ | Spe.↑ | Eff.↑ | Gen.↑ | Spe.↑ | Flu.↑ | Consis.↑ |
| Pre-edited | | $35.42_{\pm0.30}$ | $34.17_{\pm0.30}$ | $38.02_{\pm0.27}$ | $0.49_{\pm0.07}$ | $0.44_{\pm0.05}$ | $18.09_{\pm0.24}$ | $634.84_{\pm0.12}$ | $22.06_{\pm0.08}$ |
| FT | LLaMA3-8B | $15.27_{\pm0.21}$ | $14.78_{\pm0.21}$ | $5.06_{\pm0.10}$ | $\underline{8.40}_{\pm0.28}$ | $\underline{2.54}_{\pm0.13}$ | $0.03_{\pm0.01}$ | $409.80_{\pm0.67}$ | $19.35_{\pm0.13}$ |
| MEMIT | | $0.00_{\pm0.00}$ | $0.00_{\pm0.00}$ | $0.06_{\pm0.01}$ | $0.00_{\pm0.00}$ | $0.00_{\pm0.00}$ | $0.00_{\pm0.00}$ | $318.19_{\pm0.24}$ | $4.19_{\pm0.04}$ |
| PRUNE | | $10.35_{\pm0.18}$ | $10.08_{\pm0.18}$ | $9.55_{\pm0.15}$ | $1.19_{\pm0.11}$ | $0.34_{\pm0.04}$ | $\underline{0.62}_{\pm0.03}$ | $\mathbf{618.72}_{\pm0.08}$ | $\mathbf{49.24}_{\pm0.13}$ |
| RECT | | $0.01_{\pm0.00}$ | $0.01_{\pm0.01}$ | $0.04_{\pm0.01}$ | $0.57_{\pm0.08}$ | $0.29_{\pm0.04}$ | $0.10_{\pm0.01}$ | $438.83_{\pm0.18}$ | $9.40_{\pm0.05}$ |
| AlphaEdit | | $\underline{86.64}_{\pm0.23}$ | $\underline{81.28}_{\pm0.28}$ | $\underline{28.78}_{\pm0.22}$ | $4.37_{\pm0.20}$ | $1.71_{\pm0.10}$ | $0.57_{\pm0.03}$ | $482.36_{\pm0.44}$ | $4.71_{\pm0.04}$ |
| SPHERE | | $\mathbf{90.01}_{\pm0.21}$ | $\mathbf{84.67}_{\pm0.26}$ | $\mathbf{45.40}_{\pm0.29}$ | $\mathbf{52.89}_{\pm0.50}$ | $\mathbf{32.07}_{\pm0.39}$ | $\mathbf{5.01}_{\pm0.10}$ | $\underline{551.51}_{\pm0.53}$ | $\underline{30.89}_{\pm0.13}$ |
| Pre-edited | | $35.29_{\pm0.29}$ | $34.10_{\pm0.28}$ | $38.44_{\pm0.27}$ | $0.42_{\pm0.06}$ | $0.46_{\pm0.05}$ | $15.06_{\pm0.20}$ | $624.45_{\pm0.11}$ | $23.02_{\pm0.69}$ |
| FT | Qwen2.5-7B | $4.97_{\pm0.14}$ | $4.58_{\pm0.13}$ | $4.01_{\pm0.11}$ | $15.44_{\pm0.36}$ | $4.63_{\pm0.17}$ | $1.46_{\pm0.05}$ | $214.26_{\pm0.09}$ | $3.15_{\pm0.02}$ |
| MEMIT | | $0.13_{\pm0.02}$ | $0.12_{\pm0.01}$ | $0.04_{\pm0.01}$ | $0.00_{\pm0.00}$ | $0.00_{\pm0.00}$ | $0.00_{\pm0.00}$ | $370.84_{\pm0.30}$ | $3.59_{\pm0.03}$ |
| PRUNE | | $\underline{47.93}_{\pm0.36}$ | $\underline{45.50}_{\pm0.35}$ | $\underline{39.20}_{\pm0.28}$ | $14.30_{\pm0.35}$ | $11.27_{\pm0.26}$ | $\mathbf{6.75}_{\pm0.12}$ | $\mathbf{620.74}_{\pm0.10}$ | $\mathbf{29.50}_{\pm0.08}$ |
| RECT | | $0.73_{\pm0.04}$ | $0.75_{\pm0.04}$ | $0.05_{\pm0.07}$ | $0.64_{\pm0.08}$ | $0.19_{\pm0.03}$ | $0.09_{\pm0.01}$ | $368.46_{\pm0.27}$ | $1.35_{\pm0.01}$ |
| AlphaEdit | | $42.01_{\pm0.40}$ | $39.99_{\pm0.39}$ | $13.87_{\pm0.20}$ | $\underline{43.92}_{\pm0.50}$ | $\underline{24.37}_{\pm0.36}$ | $2.32_{\pm0.06}$ | $479.83_{\pm0.77}$ | $4.67_{\pm0.07}$ |
| SPHERE | | $\mathbf{70.04}_{\pm0.36}$ | $\mathbf{65.43}_{\pm0.37}$ | $27.35_{\pm0.26}$ | $\mathbf{60.76}_{\pm0.49}$ | $\mathbf{29.24}_{\pm0.37}$ | $\underline{3.83}_{\pm0.08}$ | $\underline{612.67}_{\pm0.22}$ | $\underline{14.74}_{\pm0.07}$ |

## 5.2 Performance of Sequential Model Editing (RQ1)

Table 1 presents the results under a commonly used sequential editing setup, using 15,000 samples with 100 edits each for LLaMA3 (8B), while Qwen2.5 (7B) is restricted to 5,000 edits as further updates induce severe model collapse, where all editing methods underperformed compared to the pre-edit baseline. SPHERE is plug-and-play, and AlphaEdit (Fang et al., 2025) is adopted as the default base method in Table 1 to better illustrate its capability. Additional experiments with other base methods are presented in Section 5.5. Overall, SPHERE consistently outperforms baseline methods across nearly all metrics and base models. In particular, it achieves substantial gains in both **Efficacy** and **Generalization**, with average improvements of **24.19%** and **16.02%**, respectively, over the best baseline. It also maintains notable performance in Fluency and Consistency, indicating its ability to preserve factual accuracy while generating coherent and natural outputs.

## 5.3 Analysis of Edited Weights (RQ2)

This analysis evaluates whether SPHERE can effectively maintain the hyperspherical uniformity of edited weights. We extracted the edited weights from LLaMA3 after 15,000 sequential edits on CounterFact. As shown in Figure 4, we computed the cosine similarity between each pair of weight neurons and used heatmap to visualize the hyperspherical uniformity before and after editing. In Figure 5, t-SNE (van der Maaten & Hinton, 2008) was used to visualize the normalized neuron distribution in $W$ before and after editing. It can be seen that SPHERE **effectively preserves hyperspherical uniformity after editing, as the cosine similarity among weight neurons remains close to the original distribution**, thereby avoiding directional collapse. Moreover, the pre- and post-edited weights exhibit nearly overlapping distributions, indicating that SPHERE prevents significant shifts in weights and maintains consistency. In contrast, baseline methods such as MEMIT and AlphaEdit induce clear angular concentration in neuron directions, causing neurons to cluster in limited angular regions and significantly reducing hyperspherical directional uniformity. More results on Qwen2.5 are in Appendix E.1.

## 5.4 Evaluation of General Abilities (RQ3)

To extensively evaluate whether post-edited LLMs can preserve the general abilities, four representative tasks were adopted following Gu et al. (2024), including **Reasoning** on the GSM8K (Cobbe et al., 2021), measured by solve rate. **Natural language inference (NLI)** on the RTE (Dagan et al., 2005), measured by accuracy of two-way classification, **Open-domain QA** on the Natural Questions (Kwiatkowski et al., 2019), measured by exact match (EM) with the reference answer after

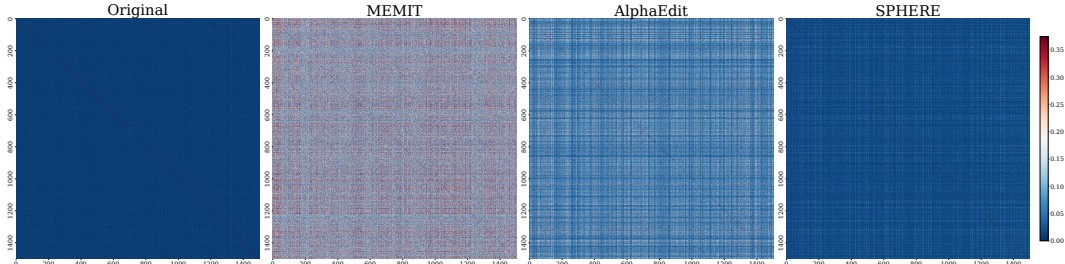

Figure 4: Cosine similarity between neurons in the updated weight matrix after 15,000 edits. Darker colors indicate lower similarity, reflecting better hyperspherical and orthogonal uniformity. SPHERE effectively preserves the weight structure, demonstrating the most stable hyperspherical uniformity.

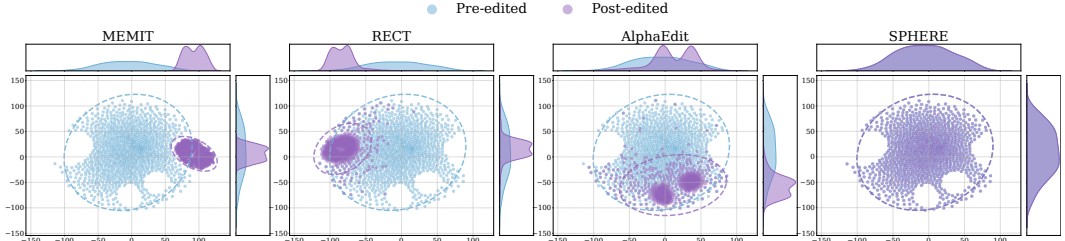

Figure 5: The t-SNE distribution of weight neurons of pre-edited and post-edited LLM after 15,000 edits using dimensionality reduction. The top and right curve graphs display the marginal distributions for two reduced dimensions, where SPHERE consistently exhibits minimal shift.

minor normalization (Chen et al., 2017; Lee et al., 2019). **Closed-domain QA** on BoolQ (Clark et al., 2019), also measured by EM. Figure 6 depicts how performance varies with the number of edited samples across four tasks. We report general performance every 1k edits up to 5k, and every 5k edits thereafter (up to 15k), providing a comprehensive view of the degradation trend. The results show that SPHERE effectively preserves the general abilities of post-edited LLMs even under extensive editing, maintaining the original model performance across all metrics after 15k edits. In contrast, LLMs edited with baseline methods rapidly lose their general abilities with all metrics approaching zero. These findings underscore the critical role of hyperspherical uniformity in safeguarding the broad abilities learned from the underlying corpus.

## 5.5 PERFORMANCE IMPROVEMENTS OF BASELINE METHODS (RQ4)

We investigated whether the sparse space projection strategy of SPHERE can serve as a general enhancement to existing methods. A single line of code from SPHERE regarding the projection was inserted into the baselines with minimal modification, and we evaluated their performance before and after integration (as detailed in Appendix D.5). Results of 3,000 sequential edits on LLaMA3 (8B) are reported in Figure 7. **SPHERE is integrated seamlessly with diverse editing methods and significantly boosts their performance.** On average, the optimized baselines achieve relative improvements of **49.05%**, **42.64%**, and **24.44%** in **Efficacy**, **Generalization**, and **Specificity**, respectively, underscoring the strong potential and broad applicability of the proposed sparse space projection as a plug-and-play enhancement for model editing. The baselines enhanced with the projection also demonstrate significantly better robustness in general abilities (see Appendix E.2).

## 5.6 ANALYSIS OF COMPUTATIONAL OVERHEAD (RQ5)

To evaluate the scalability of SPHERE, we analyze its computational cost relative to the base editing method as both the model size and the number of sequential edits increase. Specifically, we measure the runtime of the projection step introduced by SPHERE and the overall batch-editing process. Table 2 reports these runtimes

Table 2: Runtimes of SPHERE.

| Model | Edit (s) | Proj. (s) | Ratio (%) |
|---|---|---|---|
| LLaMA3-8B | 543.26 | 18.00 | 3.31 |
| Qwen2.5-7B | 535.73 | 35.95 | 6.71 |
| Qwen2.5-32B | 1656.58 | 99.60 | 6.01 |

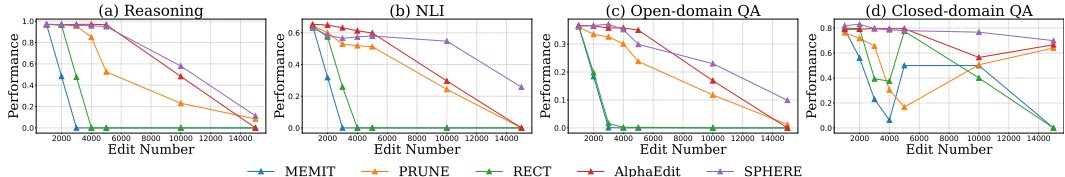

Figure 6: General ability testing of post-edited LLaMA3 (8B) on four tasks.

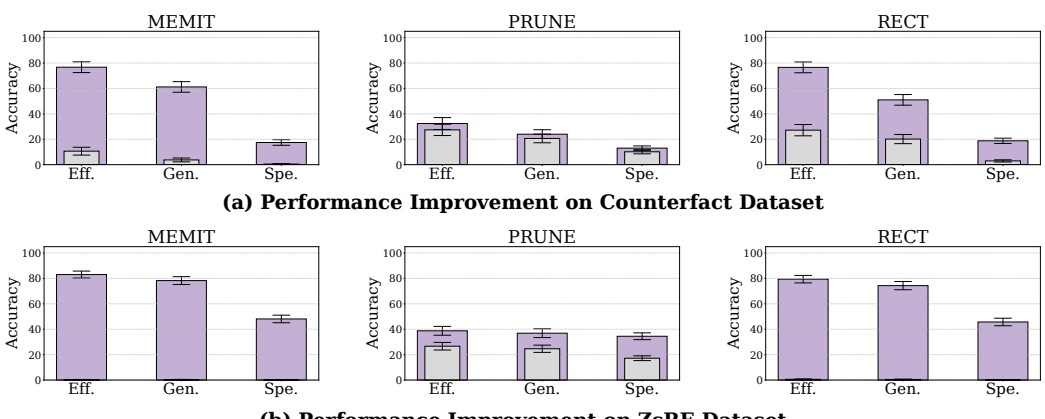

Figure 7: Performance improvements of baseline editing methods after adding a single line of code from SPHERE (i.e., sparse space projection). Gray bars denote the original baseline performance, while purple bars indicate the performance after enhancement.

together with the projection-to-total time ratio, evaluated under 3,000 sequential edits using MEMIT as the base method, with a batch size of 100. The results indicate that the projection step introduces only a minor relative cost, accounting for **3.31%** of the total editing time for LLaMA3 (8B), **6.71%** for Qwen2.5 (7B), and **6.01%** for Qwen2.5 (32B). Moreover, **the relative computational cost of SPHERE remains consistent across models of different scales within the same family**, suggesting that the additional projection operation is lightweight and does not hinder the practicality of SPHERE in large-scale batch sequential editing scenarios.

## 5.7 SENSITIVITY TO HYPERPARAMETERS (RQ6)

As analyzed in Section 3, different editing methods exhibit varying magnitudes of HE fluctuation. Since SPHERE explicitly regulates hyperspherical energy accumulation through projection, its behavior is primarily governed by the **cumulative ratio** $\eta$ and the **suppression strength** $\alpha$ (introduced in Appendix D.4.1). It is therefore important to evaluate the sensitivity of SPHERE to these two hyperparameters. To this end, we conducted an expanded ablation study that systematically varies both hyperparameters over a wide range, using MEMIT as the base method on LLaMA3 (8B) with 3,000 sequential edits, while keeping all other settings consistent with the main experiment. As shown in Figure 8, we evaluate editing performance and visualize the results using a heatmap to illustrate the performance

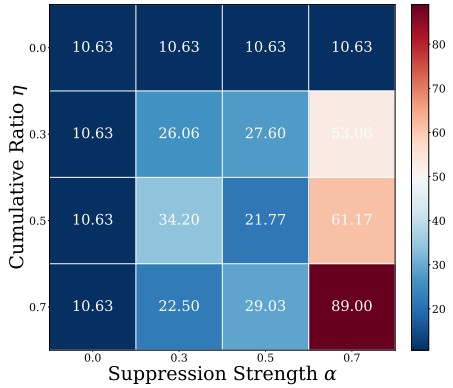

Figure 8: Sensitivity of SPHERE.

variation before and after applying SPHERE under different combinations of $(\eta, \alpha)$. Although performance fluctuations are observed under different hyperparameter settings, **across all configurations, SPHERE consistently improves the performance of its original counterpart**, demonstrating that SPHERE remains effective and robust over a wide range of hyperparameter choices. Practitioners may therefore tune these hyperparameters to suit their specific baseline or use case.

## 6 RELATED WORK

**Model Editing Methods.** From the perspective of whether model parameters are modified, existing approaches can be broadly categorized into *parameter-modifying* (Mitchell et al., 2022; Meng et al., 2023; Ma et al., 2025; Fang et al., 2025), which directly adjust a small subset of model parameters, and *parameter-preserving* (Zheng et al., 2023; Yu et al., 2024; Hartvigsen et al., 2023), which integrate auxiliary modules without altering the original weights. In this work, we focus on *parameter-modifying methods* which typically employs meta-learning or locating-then-editing strategies (Zhang et al., 2024b). Representative works of meta-learning include KE (Cao et al., 2021) and MEND (Mitchell et al., 2022), which leverage hypernetworks to generate parameter updates. Locate-then-edit methods, such as ROME (Meng et al., 2022) and MEMIT (Meng et al., 2023), prioritize pinpointing the knowledge's storage location before making targeted edits. Recent extensions like RECT (Gu et al., 2024) and PRUNE (Ma et al., 2025) mitigate degradation of general capabilities of LLMs by better constraining edit complexity via sparsity and condition number. Recently, AlphaEdit (Fang et al., 2025) further generalizes this paradigm by projecting the perturbation into the nullspace of the previous knowledge set.

**Learning with Hyperspherical Uniformity.** Early studies (Xie et al., 2017a; Rodríguez et al., 2017; Xie et al., 2017b; Cogswell et al., 2016) sought to improve the generalization capacity of neural networks by reducing redundancy through diversification, as rigorously analyzed in (Xie et al., 2016). Although these works examined angular diversity, they largely neglected the notion of global equidistribution of embeddings on the hypersphere. In contrast to orthogonality, where perpendicular vectors are defined to be diverse, hyperspherical uniformity promotes embeddings that are maximally separated in angle, thereby encouraging uniform distribution across the hypersphere (Liu et al., 2021; 2018). More recently, Smerkous et al. (2024) enhancing training stability by incorporating centered kernel alignment into hyperspherical energy, enhancing training stability by addressing the lack of permutation invariance inherent in naive similarity metrics.

## 7 CONCLUSION

In this work, we demonstrated that hyperspherical uniformity is a critical factor in stabilizing sequential editing for LLMs, supported by both empirical evidence and rigorous theoretical proof. Motivated by this insight, we propose SPHERE, a regularization strategy that preserves hyperspherical uniformity by projecting updates onto a space complementary to the principal directions of pretrained weights. Extensive evaluations on LLaMA3 (8B) and Qwen2.5 (7B) across multiple editing datasets and downstream tasks confirm that SPHERE not only enhances editing performance by 16.41% over the strongest baseline but also more faithfully preserves weight geometry and general abilities of models. Furthermore, when applied as a plug-and-play enhancement, it yields an additional average improvement of 38.71% across existing methods. Collectively, our findings establish SPHERE as both theoretically grounded and empirically effective, providing a principled and scalable solution for reliable large-scale model editing.

### ETHICS STATEMENT

SPHERE significantly enhances the reliability of large-scale sequential model editing by preserving hyperspherical uniformity, which makes it a valuable way for updating and managing knowledge in real-world applications where long-term stability is essential. At the same time, the ability to directly alter stored knowledge in LLMs carries inherent risks, including the potential introduction of bias or harmful information. To address these concerns, we strongly recommend rigorous validation procedures, transparent reporting, and strict oversight when deploying such techniques. While the core motivation of SPHERE is positive, aiming to facilitate efficient and trustworthy updates of LLMs, we emphasize that its use must remain responsible and cautious to ensure ethical outcomes.

We used LLMs to assist with improving grammar, clarity, and wording in parts of this work. The use of LLMs was limited to language refinement, with all ideas, analyses, and conclusions solely developed by the authors. We restate this announcement in Appendix A.

ACKNOWLEDGMENTS

This research is based upon work supported by NSF CAREER #2339766 and an Amazon AGI Research Award. We would like to express gratitude to the UCLANLP group members for their valuable feedback.

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

## A  USAGE OF LLMS

Throughout the preparation of this manuscript, we used LLMs to assist with improving grammar, clarity, and wording in parts of this work. The use of LLMs was limited to language refinement, with all ideas, analyses, and conclusions solely developed by the authors.

## B  PRELIMINARIES OF MODEL EDITING

Model editing aims to refine a pre-trained model by applying one or more edits, where each edit replaces a factual association $(s, r, o)$ with new knowledge $(s, r, o^*)$ (Yang et al., 2024b; Li et al., 2025). After editing, the model is expected to recall the updated object $o^*$ when given a natural language prompt $p(s, r)$, such as "The President of the United States is" (Zhang et al., 2024a).

To achieve this, locating-and-editing methods have been proposed for effective model updates (Yang et al., 2024a). These methods typically follow three steps (Jiang et al., 2025):

**Step 1: Locating Influential Layers.**  The first step is to identify the specific FFN layers that encode the target knowledge using causal tracing (Meng et al., 2022). This method involves injecting Gaussian noise into the hidden states and progressively restoring them to their original values. By analyzing the degree to which the original output recovers, the influential layers can be pinpointed as the targets for editing.

**Step 2: Acquiring the Expected Output.**  The second step aims to obtain the desired output of the critical layers identified in Step 1. Following the key–value theory, the key $k$, which encodes $(s, r)$, is processed through the output weights $W_{\text{out}}^l$ to produce the original value $v$ encoding $o$. Formally,

$$k \triangleq \sigma\big(W_{\text{in}}^l \gamma(h^{l-1} + \alpha^l)\big), \qquad v \triangleq m^l = W_{\text{out}}^l k. \tag{14}$$

To perform editing, $v$ is expected to be replaced with a new value $v^*$ encoding $o^*$. To this end, current methods typically use gradient descent on $\Delta W$, maximizing the probability that the model outputs the word associated with $o^*$ (Meng et al., 2023). The optimization objective is as follows:

$$v^* = v + \arg \min_{\Delta W^l} \Big( -\log \mathbb{P}_{f_{W_{\text{out}}}^l (m^l += \Delta W^l)} \big[ o^* \mid (s, r) \big] \Big), \tag{15}$$

where $f_{W_{\text{out}}}^l (m^l += \Delta W)$ represents the original model with $m^l$ updated to $m^l + \Delta W$.

**Step 3: Updating $W_{\text{out}}^l$.**  This step aims to update the parameters $W_{\text{out}}^l$. It includes a factual set $\{K_1, V_1\}$ containing $u$ new associations, while preserving the set $\{K_0, V_0\}$ containing $n$ original associations. Specifically,

$$\begin{aligned}
K_0 &= [\, k_1 \ k_2 \ \cdots \ k_n \,], \quad V_0 = [\, v_1 \ v_2 \ \cdots \ v_n \,], \\
K_1 &= [\, k_{n+1} \ k_{n+2} \ \cdots \ k_{n+u} \,], \quad V_1 = [\, v_{n+1}^* \ v_{n+2}^* \ \cdots \ v_{n+u}^* \,]
\end{aligned} \tag{16}$$

where $k$ and $v$ are defined in Eqn. 14 and their subscripts represent the index of the knowledge. Based on these, the objective can be defined as:

$$\tilde{W}_{\text{out}}^l \triangleq \arg \min_{\hat{W}} \left( \sum_{i=1}^{n} \big\| \hat{W} k_i - v_i \big\|^2 + \sum_{i=n+1}^{n+u} \big\| \hat{W} k_i - v_i^* \big\|^2 \right). \tag{17}$$

By applying the normal equation, its closed-form solution can be derived:

$$\tilde{W}_{\text{out}}^l = \big( M_1 - W_{\text{out}}^l K_1 \big) K_1^\top \big( K_0 K_0^\top + K_1 K_1^\top \big)^{-1} + W_{\text{out}}^l. \tag{18}$$

In practice, model editing methods often update parameters across multiple layers to improve effectiveness. For more details, see (Meng et al., 2023).

# C THEORETICAL PROOFS

## C.1 PROOF OF CORRELATION BETWEEN HYPERSPHERICAL ENERGY AND EDITING STABILITY

**Objective for Preserving Original Knowledge** We begin by assuming that the original knowledge base can be expressed as $\{\boldsymbol{k}_i,\ \boldsymbol{v}_i\}$ $i = 1, \ldots, N,$ $\boldsymbol{k}_i \in \mathbb{R}^p,\ \boldsymbol{v}_i \in \mathbb{R}^q$, while the new knowledge base is given by $\{\tilde{\boldsymbol{k}}_i,\ \tilde{\boldsymbol{v}}_i\}$ $i = 1, \ldots, N,$ $\tilde{\boldsymbol{k}}_i \in \mathbb{R}^p, \tilde{\boldsymbol{v}}_i \in \mathbb{R}^q$

And the knowledge mapping is governed by the weight matrix $W \in \mathbb{R}^{p \times q}$ such that

$$\boldsymbol{W}\boldsymbol{k}_i = \boldsymbol{v}_i, \qquad (\boldsymbol{W} + \Delta\boldsymbol{W})\tilde{\boldsymbol{k}}_i = \tilde{\boldsymbol{v}}_i. \tag{19}$$

When analyzing the destroy to the original knowledge set, there is shift brought by the perturbation $\Delta\boldsymbol{W}$ which can be expressed as:

$$(\boldsymbol{W} + \Delta\boldsymbol{W})\boldsymbol{k}_i = \boldsymbol{v}_i + \Delta\boldsymbol{W}\boldsymbol{k}_i \tag{20}$$

where we define $\Delta\boldsymbol{v}_i = \Delta\boldsymbol{W}\boldsymbol{k}_i$ as the destroy to the previous knowledge set. In addition, if $\boldsymbol{k}_i \in \text{null}(\Delta\boldsymbol{W})$, i.e., in the null space of $\Delta\boldsymbol{W}$, then $\Delta\boldsymbol{W}\boldsymbol{k}_i = \Delta\boldsymbol{v}_i = 0$.

The corresponding objective is thus to minimize the perturbation magnitude, given by

$$\min \frac{1}{N} \sum_{i=1}^{N} \|\Delta\boldsymbol{v}_i\| \quad \equiv \quad \min \frac{1}{N} \sum_{i=1}^{N} \|\Delta\boldsymbol{W}\boldsymbol{k}_i\|. \tag{21}$$

To make this tractable, assume that each input vector $\boldsymbol{k}_i$ can be approximated in terms of the first $\boldsymbol{K}$ basis vectors $\{\boldsymbol{e_j}\}$ as

$$\boldsymbol{k}_i = \sum_{j=1}^{K} \boldsymbol{\alpha_j}\boldsymbol{e_j} + \varepsilon_i, \tag{22}$$

where $\varepsilon_i$ is a small noise term. If we denote

$$\Delta\boldsymbol{W} \cdot \boldsymbol{e_j} = \boldsymbol{f_j}, \qquad \varepsilon_i, \boldsymbol{e_j}, \boldsymbol{f_j} \in \mathbb{R}^q, \tag{23}$$

then the perturbation objective can be rewritten as:

$$\begin{aligned} \min \frac{1}{N} \sum_{i=1}^{N} &\left\| \Delta\boldsymbol{W} \cdot \left( \sum_{j=1}^{K} \boldsymbol{\alpha_j}\boldsymbol{e_j} + \varepsilon_i \right) \right\| \\ &\leq \frac{1}{N} \sum_{i=1}^{N} \left\| \sum_{j=1}^{K} \boldsymbol{\alpha_j}\boldsymbol{f_j} + \Delta\boldsymbol{W}\varepsilon_i \right\| \\ &\leq \frac{1}{N} \sum_{i=1}^{N} \sum_{j=1}^{K} |\boldsymbol{\alpha_j}\boldsymbol{f_j}| + \|\Delta\boldsymbol{W}\|\|\varepsilon_i\| \\ &\leq \frac{1}{N} \sum_{i=1}^{N} \sum_{j=1}^{K} |\boldsymbol{\alpha_j}| \|\boldsymbol{f_j}\| + \varepsilon_{\max} \|\boldsymbol{\Delta\boldsymbol{W}}\|. \end{aligned} \tag{24}$$

where $\varepsilon_{\max} = \max_i \|\varepsilon_i\|$. This shows that minimizing $\|f_j\|$ directly reduces the upper bound of the perturbation, and therefore enhances the stability of the editing process.

**Definitions** Let the model's weights be represented by a matrix $\boldsymbol{W} \in \mathbb{R}^{p \times q}$, whose rows are the neuron vectors $\boldsymbol{w}_1, \ldots, \boldsymbol{w}_p \in \mathbb{R}^q$. An edit or update introduces a perturbation to this matrix, denoted by $\Delta\boldsymbol{W} \in \mathbb{R}^{p \times q}$, with corresponding row-wise perturbations $\Delta\boldsymbol{w}_1, \ldots, \Delta\boldsymbol{w}_p$.

We define two key scalar quantities to measure the effects of this perturbation:

- **output perturbation ($\Delta \boldsymbol{V}$).** This quantity measures the total squared change in the model's output, aggregated over a set of $N$ input vectors $\{\boldsymbol{k}_i\}_{i=1}^{N}$.

$$\Delta \boldsymbol{V} \triangleq \sum_{i=1}^{N} \|\Delta \boldsymbol{W} \boldsymbol{k}_i\|_2^2 = \sum_{i=1}^{N} \left\| \begin{bmatrix} \Delta \boldsymbol{w}_1 \cdot \boldsymbol{k}_i \\ \vdots \\ \Delta \boldsymbol{w}_p \cdot \boldsymbol{k}_i \end{bmatrix} \right\|_2^2 = \sum_{i=1}^{N} \sum_{j=1}^{p} (\Delta \boldsymbol{w}_j \cdot \boldsymbol{k}_i)^2 \tag{25}$$

- **Change in Hyperdimensional Energy ($\Delta \boldsymbol{HE}$).** This quantity measures the change in the geometric arrangement of the neuron vectors due to the perturbation.

$$\Delta \boldsymbol{HE} \triangleq \sum_{i \neq j} \left( \frac{1}{\|\boldsymbol{w}_i - \boldsymbol{w}_j\|_2^2} - \frac{1}{\|(\boldsymbol{w}_i + \Delta \boldsymbol{w}_i) - (\boldsymbol{w}_j + \Delta \boldsymbol{w}_j)\|_2^2} \right) \tag{26}$$

**Assumptions** Our analysis relies on the following assumptions:

**Assumption 1** (Orthonormal Inputs). *The set of input vectors $\{\boldsymbol{k}_i\}_{i=1}^{q}$ is the standard orthonormal basis of $\mathbb{R}^q$.*

Under this assumption, the output perturbation simplifies to the squared Frobenius norm of the perturbation matrix:

$$\Delta \boldsymbol{V} = \sum_{i=1}^{q} \sum_{j=1}^{p} (\Delta \boldsymbol{w}_j \cdot \boldsymbol{k}_i)^2 = \sum_{j=1}^{p} \sum_{i=1}^{q} (\Delta \boldsymbol{w}_{j,i})^2 = \sum_{j=1}^{p} \|\Delta \boldsymbol{w}_j\|_2^2 = \|\Delta \boldsymbol{W}\|_F^2$$

**Assumption 2** (Small Perturbations). *The perturbation vectors $\Delta \boldsymbol{w}_i$ are sufficiently small in norm, which justifies the use of a first-order Taylor expansion to approximate the change in HE.*

We can now state the relationship between the change in HE and the output perturbation energy.

**Theorem 2** (Upper Bound on HE Change). *Under Assumptions 1 and 2, the absolute change in Hyperdimensional Energy, $|\Delta \boldsymbol{HE}|$, is upper-bounded by the square root of the output perturbation, $\sqrt{\Delta \boldsymbol{V}}$, as follows:*

$$|\Delta \boldsymbol{HE}| \leq K \sqrt{\Delta \boldsymbol{V}} \tag{27}$$

*where $K$ is a constant determined by the geometry of the original weight matrix $\boldsymbol{W}$:*

$$K = 4 \sqrt{\sum_{k=1}^{p} \left( \sum_{j \neq k} \|\boldsymbol{w}_k - \boldsymbol{w}_j\|^{-3} \right)^2}$$

*Proof.* Let $\boldsymbol{p}_{ij} = \boldsymbol{w}_i - \boldsymbol{w}_j$ and $\Delta \boldsymbol{p}_{ij} = \Delta \boldsymbol{w}_i - \Delta \boldsymbol{w}_j$. The change in HE is $\Delta \boldsymbol{HE} = \sum_{i \neq j} (\|\boldsymbol{p}_{ij}\|^{-2} - \|\boldsymbol{p}_{ij} + \Delta \boldsymbol{p}_{ij}\|^{-2})$. Using a first-order Taylor expansion for $f(\boldsymbol{x}) = \|\boldsymbol{x}\|^{-2}$ around $\boldsymbol{p}_{ij}$, we have:

$$\|\boldsymbol{p}_{ij} + \Delta \boldsymbol{p}_{ij}\|^{-2} \approx \|\boldsymbol{p}_{ij}\|^{-2} - 2\|\boldsymbol{p}_{ij}\|^{-4} (\boldsymbol{p}_{ij} \cdot \Delta \boldsymbol{p}_{ij})$$

Substituting this into the expression for $\Delta \boldsymbol{HE}$:

$$\Delta \boldsymbol{HE} \approx \sum_{i \neq j} \left( \|\boldsymbol{p}_{ij}\|^{-2} - \left( \|\boldsymbol{p}_{ij}\|^{-2} - 2\|\boldsymbol{p}_{ij}\|^{-4} (\boldsymbol{p}_{ij} \cdot \Delta \boldsymbol{p}_{ij}) \right) \right)$$

$$= \sum_{i \neq j} 2\|\boldsymbol{p}_{ij}\|^{-4} (\boldsymbol{p}_{ij} \cdot \Delta \boldsymbol{p}_{ij})$$

We bound the absolute value of this approximation:

$$|\Delta \boldsymbol{HE}| \approx \left| \sum_{i \neq j} 2\|\boldsymbol{w}_i - \boldsymbol{w}_j\|^{-4} ((\boldsymbol{w}_i - \boldsymbol{w}_j) \cdot (\Delta \boldsymbol{w}_i - \Delta \boldsymbol{w}_j)) \right|$$

$$\leq \sum_{i \neq j} 2\|\boldsymbol{w}_i - \boldsymbol{w}_j\|^{-3} \|\Delta \boldsymbol{w}_i - \Delta \boldsymbol{w}_j\| \qquad \text{(by Cauchy-Schwarz)}$$

$$\leq \sum_{i \neq j} 2\|\boldsymbol{w}_i - \boldsymbol{w}_j\|^{-3} (\|\Delta \boldsymbol{w}_i\| + \|\Delta \boldsymbol{w}_j\|) \qquad \text{(by Triangle Inequality)}$$

By re-indexing the sum to group terms by $\|\Delta \boldsymbol{w}_k\|$:

$$|\Delta \boldsymbol{H}\boldsymbol{E}| \leq \sum_{k=1}^{p} \left( \sum_{j \neq k} 2\|\boldsymbol{w}_k - \boldsymbol{w}_j\|^{-3} + \sum_{i \neq k} 2\|\boldsymbol{w}_i - \boldsymbol{w}_k\|^{-3} \right) \|\Delta \boldsymbol{w}_k\|$$

$$= \sum_{k=1}^{p} \left( 4 \sum_{j \neq k} \|\boldsymbol{w}_k - \boldsymbol{w}_j\|^{-3} \right) \|\Delta \boldsymbol{w}_k\|$$

Applying the Cauchy-Schwarz inequality to this final sum (viewed as a dot product in $\mathbb{R}^p$):

$$|\Delta \boldsymbol{H}\boldsymbol{E}| \leq \sqrt{\sum_{k=1}^{p} \left( 4 \sum_{j \neq k} \|\boldsymbol{w}_k - \boldsymbol{w}_j\|^{-3} \right)^2} \cdot \sqrt{\sum_{k=1}^{p} \|\Delta \boldsymbol{w}_k\|^2}$$

$$= K \cdot \sqrt{\sum_{k=1}^{p} \|\Delta \boldsymbol{w}_k\|^2}$$

From Assumption 1, we know $\sum_{k=1}^{p} \|\Delta \boldsymbol{w}_k\|^2 = \Delta \boldsymbol{V}$. Therefore:

$$|\Delta \boldsymbol{H}\boldsymbol{E}| \leq K\sqrt{\Delta \boldsymbol{V}}$$

$\square$

## C.2 Proof of Correlation Between Sparse Space Projection and Hyperspherical Energy

**Lemma 1.** *For any vector $\boldsymbol{x} \in \mathbb{R}^d$ and a small perturbation $\Delta \boldsymbol{x} \in \mathbb{R}^d$, the first-order Taylor expansion of the function $g(\boldsymbol{x}) = \|\boldsymbol{x}\|_2^{-s}$ is:*

$$g(\boldsymbol{x} + \Delta \boldsymbol{x}) \approx g(\boldsymbol{x}) + \nabla g(\boldsymbol{x})^T \Delta \boldsymbol{x} = \|\boldsymbol{x}\|_2^{-s} - s\|\boldsymbol{x}\|_2^{-s-2} x^T \Delta \boldsymbol{x}. \tag{28}$$

We have

$$g(\boldsymbol{x}) = \left( \sum_k \boldsymbol{x}_k^2 \right)^{-s/2}. \tag{29}$$

The partial derivative with respect to $x_l$ is:

$$\frac{\partial g}{\partial \boldsymbol{x}_l} = -\frac{s}{2} \left( \sum_k \boldsymbol{x}_k^2 \right)^{-s/2-1} \cdot (2\boldsymbol{x}_l) = -s\|\boldsymbol{x}\|_2^{-s-2} \boldsymbol{x}_l. \tag{30}$$

Thus, the gradient vector is

$$\nabla g(\boldsymbol{x}) = -s\|\boldsymbol{x}\|_2^{-s-2} \boldsymbol{x}. \tag{31}$$

Substituting into the first-order Taylor expansion

$$g(\boldsymbol{x} + \Delta \boldsymbol{x}) \approx g(\boldsymbol{x}) + \nabla g(\boldsymbol{x})^T \Delta \boldsymbol{x} \tag{32}$$

completes the proof.

**Theorem 3.** *The magnitude of $|\Delta \boldsymbol{H}\boldsymbol{E}|$ is bounded above by a constant-weighted sum of all neuron perturbation norms:*

$$|\Delta \boldsymbol{H}\boldsymbol{E}| \leq \sum_{k=1}^{p} C_k \|\Delta \boldsymbol{w}_k\|_2, \tag{33}$$

*where*

$$C_k = s \sum_{j \neq k} \|\boldsymbol{w}_k - \boldsymbol{w}_j\|_2^{-s-1} \tag{34}$$

*is a constant that depends only on the original weight matrix $\boldsymbol{W}$.*

Consider each term in $\Delta \boldsymbol{HE}$. Let

$$\boldsymbol{p}_{ij} = \boldsymbol{w}_i - \boldsymbol{w}_j, \quad \Delta \boldsymbol{p}_{ij} = (\boldsymbol{w}'_i - \boldsymbol{w}'_j) - \boldsymbol{p}_{ij} = \Delta \boldsymbol{w}_i - \Delta \boldsymbol{w}_j. \tag{35}$$

Then

$$\Delta \boldsymbol{HE} = \sum_{i<j} \left( \|\boldsymbol{p}_{ij}\|_2^{-s} - \|\boldsymbol{p}_{ij} + \Delta \boldsymbol{p}_{ij}\|_2^{-s} \right). \tag{36}$$

By Lemma 1:

$$\|\boldsymbol{p}_{ij} + \Delta \boldsymbol{p}_{ij}\|_2^{-s} \approx \|\boldsymbol{p}_{ij}\|_2^{-s} - s\|\boldsymbol{p}_{ij}\|_2^{-s-2} \boldsymbol{p}_{ij}^T \Delta \boldsymbol{p}_{ij}. \tag{37}$$

Substituting into the expression for $\Delta HE$:

$$\Delta \boldsymbol{HE} \approx \sum_{i<j} s\|\boldsymbol{p}_{ij}\|_2^{-s-2} \boldsymbol{p}_{ij}^T \Delta \boldsymbol{p}_{ij}. \tag{38}$$

To obtain a rigorous bound, apply the mean value theorem. For $g(\boldsymbol{x}) = \|\boldsymbol{x}\|_2^{-s}$, there exists $\xi_{ij}$ between $\boldsymbol{p}_{ij}$ and $\boldsymbol{p}_{ij} + \Delta \boldsymbol{p}_{ij}$ such that

$$g(\boldsymbol{p}_{ij} + \Delta \boldsymbol{p}_{ij}) - g(\boldsymbol{p}_{ij}) = \nabla g(\xi_{ij})^T \Delta \boldsymbol{p}_{ij}. \tag{39}$$

Taking absolute values and applying the Cauchy–Schwarz inequality:

$$|g(\boldsymbol{p}_{ij} + \Delta \boldsymbol{p}_{ij}) - g(\boldsymbol{p}_{ij})| \leq \|\nabla g(\xi_{ij})\|_2 \cdot \|\Delta \boldsymbol{p}_{ij}\|_2. \tag{40}$$

Since

$$\nabla g(\boldsymbol{x}) = -s\|\boldsymbol{x}\|_2^{-s-2} \boldsymbol{x}, \tag{41}$$

its norm is

$$\|\nabla g(\boldsymbol{x})\|_2 = s\|\boldsymbol{x}\|_2^{-s-1}. \tag{42}$$

Assuming small perturbations, $\xi_{ij} \approx \boldsymbol{p}_{ij}$, giving

$$|g(\boldsymbol{p}_{ij} + \Delta \boldsymbol{p}_{ij}) - g(\boldsymbol{p}_{ij})| \approx s\|\boldsymbol{p}_{ij}\|_2^{-s-1} \|\Delta \boldsymbol{p}_{ij}\|_2. \tag{43}$$

Thus,

$$|\Delta HE| \leq \sum_{i<j} s\|\boldsymbol{w}_i - \boldsymbol{w}_j\|_2^{-s-1} \|\Delta \boldsymbol{p}_{ij}\|_2. \tag{44}$$

Applying the triangle inequality:

$$\|\Delta \boldsymbol{p}_{ij}\|_2 = \|\Delta \boldsymbol{w}_i - \Delta \boldsymbol{w}_j\|_2 \leq \|\Delta \boldsymbol{w}_i\|_2 + \|\Delta \boldsymbol{w}_j\|_2. \tag{45}$$

Therefore,

$$|\Delta \boldsymbol{HE}| \leq \sum_{i<j} s\|\boldsymbol{w}_i - \boldsymbol{w}_j\|_2^{-s-1} (\|\Delta \boldsymbol{w}_i\|_2 + \|\Delta \boldsymbol{w}_j\|_2). \tag{46}$$

Rearranging terms with respect to each $\|\Delta \boldsymbol{w}_k\|_2$, we obtain:

$$|\Delta \boldsymbol{HE}| \leq \sum_{k=1}^{p} \|\Delta \boldsymbol{w}_k\|_2 \left( s \sum_{j \neq k} \|\boldsymbol{w}_k - \boldsymbol{w}_j\|_2^{-s-1} \right). \tag{47}$$

Defining

$$C_k = s \sum_{j \neq k} \|\boldsymbol{w}_k - \boldsymbol{w}_j\|_2^{-s-1}, \tag{48}$$

we conclude

$$|\Delta \boldsymbol{HE}| \leq \sum_{k=1}^{p} C_k \|\Delta \boldsymbol{w}_k\|_2. \tag{49}$$

**Conclusion of Theorem 1.** The magnitude of $|\Delta \boldsymbol{HE}|$ is constrained by the weighted sum of neuron perturbation norms. To reduce $|\Delta \boldsymbol{HE}|$, an effective approach is to minimize each $\|\Delta \boldsymbol{w}_k\|_2$.

**Theorem 4.** *The SPHERE projection operation reduces (or preserves) the $\ell_2$-norm of perturbation vectors:*

$$\|\Delta \boldsymbol{w}_{i,SPHERE}\|_2 \leq \|\Delta \boldsymbol{w}_i\|_2. \tag{50}$$

Compute the squared $\ell_2$-norm:

$$\|\Delta\boldsymbol{w}_{i,\text{SPHERE}}\|_2^2 = \|\Delta\boldsymbol{w}_i\boldsymbol{P}_\perp\|_2^2 = (\Delta\boldsymbol{w}_i\boldsymbol{P}_\perp)(\Delta\boldsymbol{w}_i\boldsymbol{P}_\perp)^T. \tag{51}$$

Using $(AB)^T = B^T A^T$:

$$\|\Delta\boldsymbol{w}_{i,\text{SPHERE}}\|_2^2 = \Delta\boldsymbol{w}_i\boldsymbol{P}_\perp\boldsymbol{P}_\perp^T\Delta\boldsymbol{w}_i^T. \tag{52}$$

The projection matrix $\boldsymbol{P}_\perp$ satisfies two key properties:

- **Symmetry:** $\boldsymbol{P}_\perp^T = \boldsymbol{P}_\perp$.
- **Idempotence:** $\boldsymbol{P}_\perp^2 = \boldsymbol{P}_\perp$.

Thus,

$$\|\Delta\boldsymbol{w}_{i,\text{SPHERE}}\|_2^2 = \Delta\boldsymbol{w}_i\boldsymbol{P}_\perp^2\Delta\boldsymbol{w}_i^T = \Delta\boldsymbol{w}_i\boldsymbol{P}_\perp\Delta\boldsymbol{w}_i^T. \tag{53}$$

Substituting $\boldsymbol{P}_\perp = I - \boldsymbol{U}\boldsymbol{U}^T$:

$$\|\Delta\boldsymbol{w}_{i,\text{SPHERE}}\|_2^2 = \Delta\boldsymbol{w}_i(I - \boldsymbol{U}\boldsymbol{U}^T)\Delta\boldsymbol{w}_i^T = \|\Delta\boldsymbol{w}_i\|_2^2 - \|\Delta\boldsymbol{w}_i\boldsymbol{U}\|_2^2. \tag{54}$$

Since

$$\|\Delta\boldsymbol{w}_i\boldsymbol{U}\|_2^2 \geq 0, \tag{55}$$

we conclude

$$\|\Delta\boldsymbol{w}_{i,\text{SPHERE}}\|_2^2 \leq \|\Delta\boldsymbol{w}_i\|_2^2. \tag{56}$$

Taking square roots:

$$\|\Delta\boldsymbol{w}_{i,\text{SPHERE}}\|_2 \leq \|\Delta\boldsymbol{w}_i\|_2. \tag{57}$$

Equality holds iff

$$\Delta\boldsymbol{w}_i\boldsymbol{U} = 0, \tag{58}$$

i.e., $\Delta\boldsymbol{w}_i$ is orthogonal to all basis vectors of the principal subspace $\boldsymbol{U}$. In this case, $\Delta\boldsymbol{w}_i$ already lies in the sparse subspace.

## D  EXPERIMENTAL SETUP

### D.1  DATASETS

Here, we provide a detailed introduction to the datasets used in this paper:

- **Counterfact** (Meng et al., 2022) is a more challenging dataset that contrasts counterfactual with factual statements, initially scoring lower for Counterfact. It constructs out-of-scope data by replacing the subject entity with approximate entities sharing the same predicate. The Counterfact dataset has similar metrics to ZsRE for evaluating efficacy, generalization, and specificity. Additionally, Counterfact includes multiple generation prompts with the same meaning as the original prompt to test the quality of generated text, specifically focusing on fluency and consistency.

- **ZsRE** (Levy et al., 2017) is a question answering (QA) dataset that uses questions generated through back-translation as equivalent neighbors. Following previous work, natural questions are used as out-of-scope data to evaluate locality. Each sample in ZsRE includes a subject string and answers as the editing targets to assess editing success, along with the rephrased question for generalization evaluation and the locality question for evaluating specificity.

### D.2  EVALUATION METRICS

Now we introduce the evaluation metrics for the ZsRE and Counterfact datasets, respectively.

### D.2.1 METRICS FOR ZSRE

Following the previous work (Mitchell et al., 2022; Meng et al., 2022; 2023), this section defines each ZsRE metric given a LLM $f_\theta$, a knowledge fact prompt $(s_i, r_i)$, an edited target output $o_i$, and the model's original output $o_i^c$:

- **Efficacy:** Efficacy is calculated as the average top-1 accuracy on the edit samples:

$$\mathbb{E}_i\Big\{o_i = \arg\max_o \mathbb{P}_{f_\theta}(o \mid (s_i, r_i))\Big\}. \tag{59}$$

- **Generalization:** Generalization measures the model's performance on equivalent prompts of $(s_i, r_i)$, such as rephrased statements $N((s_i, r_i))$. This is evaluated by the average top-1 accuracy on these $N((s_i, r_i))$:

$$\mathbb{E}_i\Big\{o_i = \arg\max_o \mathbb{P}_{f_\theta}(o \mid N((s_i, r_i)))\Big\}. \tag{60}$$

- **Specificity:** Specificity ensures that the editing does not affect samples unrelated to the edit cases $O(s_i, r_i)$. This is evaluated by the top-1 accuracy of predictions that remain unchanged:

$$\mathbb{E}_i\Big\{o_i^c = \arg\max_o \mathbb{P}_{f_\theta}(o \mid O((s_i, r_i)))\Big\}. \tag{61}$$

### D.2.2 METRICS FOR COUNTERFACT

Following previous work (Meng et al., 2022; 2023), this section defines the Counterfact metrics given a language model $f_\theta$, a knowledge fact prompt $(s_i, r_i)$, an edited target output $o_i$, and the model's original output $o_i^c$. However, for rigorous evaluation, we adopt the **average top-1 accuracy** as the metric for this dataset, which is used to assess Efficacy, Generalization, and Specificity.

- **Efficacy (efficacy success):** Efficacy is calculated as the average top-1 accuracy on the edit samples:

$$\mathbb{E}_i\Big\{o_i = \arg\max_o \mathbb{P}_{f_\theta}(o \mid (s_i, r_i))\Big\}. \tag{62}$$

- **Generalization (paraphrase success):** Generalization measures the model's performance on equivalent prompts of $(s_i, r_i)$, such as rephrased statements $N((s_i, r_i))$. This is evaluated by the average top-1 accuracy on these $N((s_i, r_i))$:

$$\mathbb{E}_i\Big\{o_i = \arg\max_o \mathbb{P}_{f_\theta}(o \mid N((s_i, r_i)))\Big\}. \tag{63}$$

- **Specificity (neighborhood success):** Specificity ensures that the editing does not affect samples unrelated to the edit cases $O(s_i, r_i)$. This is evaluated by the top-1 accuracy of predictions that remain unchanged:

$$\mathbb{E}_i\Big\{o_i^c = \arg\max_o \mathbb{P}_{f_\theta}(o \mid O((s_i, r_i)))\Big\}. \tag{64}$$

- **Fluency (generation entropy):** Measure for excessive repetition in model outputs. It uses the entropy of n-gram distributions:

$$-\frac{2}{3}\sum_k g_2(k)\log_2 g_2(k) + \frac{4}{3}\sum_k g_3(k)\log_2 g_3(k), \tag{65}$$

where $g_n(\cdot)$ is the n-gram frequency distribution.

- **Consistency (reference score):** The consistency of the model's outputs is evaluated by giving the model $f_\theta$ a subject $s$ and computing the cosine similarity between the TF-IDF vectors of the model-generated text and a reference Wikipedia text about $o$.

### D.3 BASELINES

We introduce the five baseline models employed in this study. **For the hyperparameter settings of the baseline methods, except those mentioned in Appendix D.4, we follow the original code provided in the respective papers for reproduction**.

- **MEMIT** is a scalable multi-layer editing algorithm designed to insert new factual memories into transformer-based language models. Extending ROME, MEMIT targets transformer module weights that mediate factual recall, allowing efficient updates of thousands of associations with improved scalability.

- **PRUNE** preserves the general abilities of LLMs during sequential editing by constraining numerical sensitivity. It addresses performance degradation from repeated edits by applying condition number restraints to the edited matrix, thereby limiting harmful perturbations to stored knowledge and ensuring edits can be made without compromising overall model capability.

- **RECT** mitigates unintended side effects of model editing on general reasoning and question answering. It regularizes weight updates during editing to prevent excessive alterations that cause overfitting, thereby maintaining strong editing performance while preserving the model's broader generalization abilities.

- **AlphaEdit** introduces a sequential editing framework that leverages null-space projection to constrain parameter updates. By projecting edits into the null space of unrelated knowledge, AlphaEdit reduces interference with pre-existing capabilities and improves stability under sequential edits. This design enables efficient large-scale editing with enhanced robustness and generalization compared to prior approaches.

### D.4 IMPLEMENTATION DETAILS

Our implementation of SPHERE with Llama3 (8B) and Qwen2.5 (7B) follows the configurations outlined in MEMIT (Meng et al., 2023). Specifically, we edit critical layers [4, 5, 6, 7, 8], with the hyperparameters $\eta$ set to 0.5 and $\alpha$ set to 0.5 (see Appendix D.4.1). During hidden representation updates of the critical layer, we perform 25 optimization steps. The learning rate were set to 0.1 for Llama3 (8B) and 0.5 for Qwen2.5 (7B), respectively. All experiments are conducted on eight A800 (80GB) GPUs. The LLMs are loaded using HuggingFace Transformers (Wolf et al., 2019).

#### D.4.1 CUMULATIVE RATIO $\eta$ AND SUPPRESSION STRENGTH $\alpha$

We next provide details of two important hyperparameters in our sparse space projection: the cumulative ratio $\eta$ and the suppression strength $\alpha$, together with the values used in our experiments.

**Cumulative Ratio $\eta$.** We define $\eta$ as the cumulative ratio used to select the top $r$ eigenvectors in Eqn. 66, corresponding to the $r$ principal directions on the unit hypersphere. Specifically, $\eta$ controls the selection of eigenvectors based on their eigenvalues $\lambda$, such that

$$\sum_{i=d-r+1}^{d} \lambda_i \ \geq \ \eta \cdot \sum_{i=1}^{d} \lambda_i.$$

In practice, we set $\eta = 0.5$ for all experiments, meaning that only the top 50% of the principal directions of the edited weights are suppressed.

$$\boldsymbol{U} = [v_{d-r+1}, \ldots, v_d] \in \mathbb{R}^{d \times r}. \tag{66}$$

**Suppression Strength $\alpha$.** We define $\alpha$ as the suppression strength in the projection, which controls the extent to which perturbation components along the principal directions $\boldsymbol{U}$ are removed, as shown in Eqn. 67. In practice, we set $\alpha = 0.5$ for projections on AlphaEdit, while using $\alpha = 0.8$ for all other methods, following the empirical findings reported in Section 3.1 (Observation 2).

$$\boldsymbol{P}_\perp = I - \alpha \boldsymbol{U}\boldsymbol{U}^\top \in \mathbb{R}^{d \times d}. \tag{67}$$

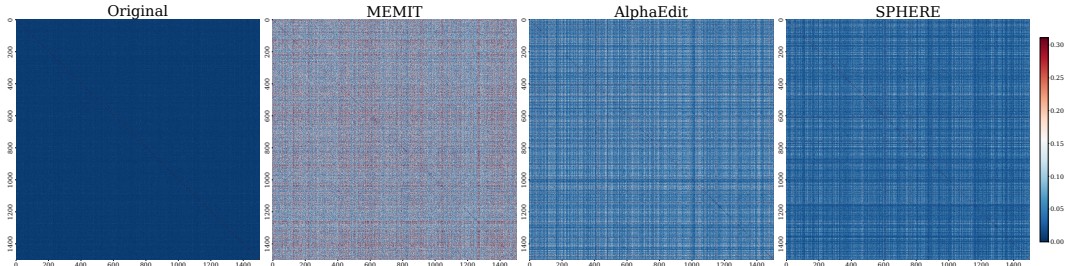

Figure 9: Cosine similarity between neurons in updated weight matrix after 5,000 edits on Qwen2.5. Darker colors indicate lower similarity, reflecting better hyperspherical and orthogonal uniformity. SPHERE effectively preserve the weight structure, demonstrating the most stable hyperspherical uniformity.

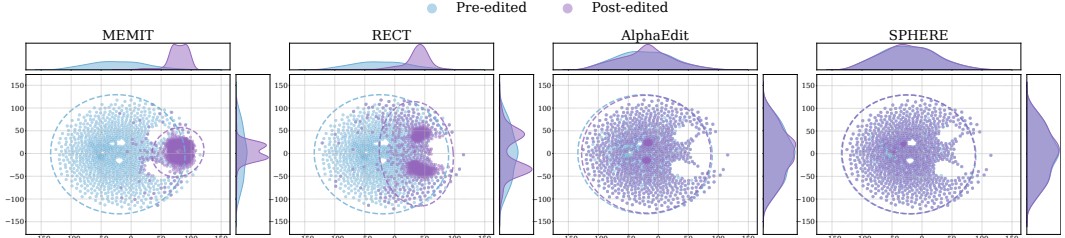

Figure 10: The distribution of weight neurons of pre-edited and post-edited Qwen2.5 after 5,000 edits using dimensionality reduction across mainstream sequential editing methods. The top and right curve graphs display the marginal distributions for two reduced dimensions, where SPHERE consistently exhibits minimal shift.

## D.5 ADDING PROJECTION IN BASELINE METHODS

We then describe the details of incorporating our projection into baseline editing methods. For illustration, we take MEMIT as an example, though the same procedure is applied to all other methods (*i.e.* FT, PRUNE, RECT, and AlphaEdit).

As introduced in Section 2.1, the editing objective can be written as:

$$\Delta \boldsymbol{W} = \arg \min_{\Delta \hat{\boldsymbol{W}}} \left( \left\| (\boldsymbol{W} + \Delta \hat{\boldsymbol{W}}) \boldsymbol{K}_1 - \boldsymbol{V}_1 \right\|^2 + \left\| (\boldsymbol{W} + \Delta \hat{\boldsymbol{W}}) \boldsymbol{K}_0 - \boldsymbol{V}_0 \right\|^2 \right). \quad (68)$$

Then, the solution for Eqn. 68 can be expressed as (Fang et al., 2025):

$$\Delta \boldsymbol{W}_{\text{MEMIT}} = \boldsymbol{R} \boldsymbol{K}_1^T \left( \boldsymbol{K}_p \boldsymbol{K}_p^T + \boldsymbol{K}_1 \boldsymbol{K}_1^T + \boldsymbol{K}_0 \boldsymbol{K}_0^T \right)^{-1}, \quad (69)$$

where $\boldsymbol{K}_p$ denotes the key and value matrices of previously updated knowledge, analogous to $\boldsymbol{K}_1$ and $\boldsymbol{V}_1$, and $\boldsymbol{R} = \boldsymbol{V}_1 - \boldsymbol{W} \boldsymbol{K}_1$.

In our sparse-space projection framework, the projection matrix does not directly participate in solving the above optimization problem. Instead, we first obtain $\Delta \boldsymbol{W}$ from the normal equation (or other solvers), and then apply the projection afterwards, as follows:

$$\hat{\boldsymbol{W}} = \boldsymbol{W} + \Delta \boldsymbol{W} \boldsymbol{P}_\perp. \quad (70)$$

This design makes the projection step modular and easily generalizable across different editing algorithms.

## E MORE EXPERIMENTAL RESULTS

### E.1 ANALYSIS OF EDITED WEIGHTS

As illustrated in Figure 9 and 10, SPHERE effectively preserves hyperspherical uniformity after editing on Qwen2.5 (7B) as well, as the cosine similarity among weight neurons remains close

to the original distribution, thereby avoiding directional collapse and maintaining its hyperspherical uniformity. Moreover, the pre- and post-edited weights exhibit more similar distributions, indicating that SPHERE prevents significant shifts in hidden representations and maintains consistency. In contrast, all other baselines induce clear angular concentration in neuron directions, causing neurons to cluster in limited angular regions and significantly reducing hyperspherical directional diversity.

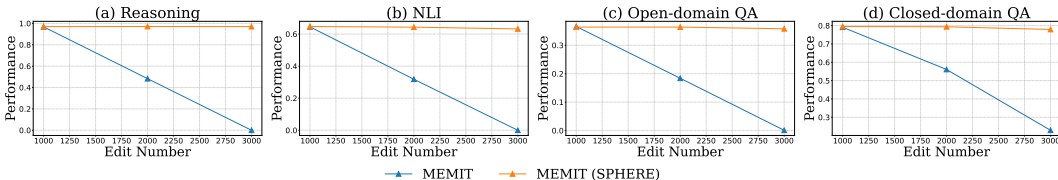

Figure 11: General ability improvements of MEMIT after incorporating SPHERE with a single line of sparse space projection code.)

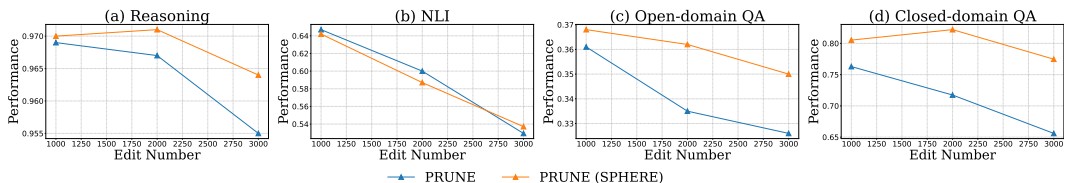

Figure 12: General ability improvements of PRUNE after incorporating SPHERE with a single line of sparse space projection code.)

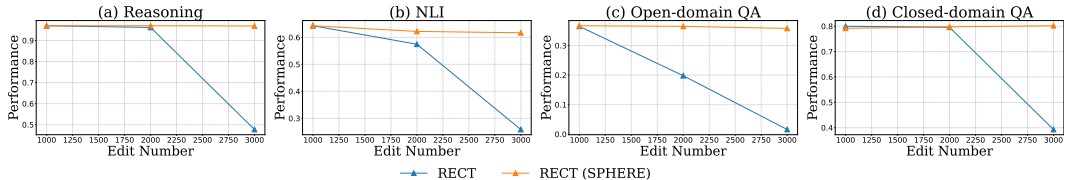

Figure 13: General ability improvements of RECT after incorporating SPHERE with a single line of sparse space projection code.)

### E.2 GENERAL ABILITY TEST ON BASELINE IMPROVEMENT

In this section, we aim to provide a comprehensive assessment of general ability performance, complementing Section 5.5, ,with results illustrated in Figure 11 for MEMIT, Figure 12 for PRUNE, and Figure 13 for RECT. Following the evaluation protocol of Gu et al. (2024), we adopt four representative tasks to measure general abilities: **Reasoning**, evaluated on GSM8K (Cobbe et al., 2021) using solve rate; **Natural Language Inference (NLI)**, evaluated on RTE (Dagan et al., 2005) using two-way classification accuracy; **Open-domain QA**, evaluated on Natural Questions (Kwiatkowski et al., 2019) using exact match (EM) against the reference answer with minor normalization (Chen et al., 2017; Lee et al., 2019); and **Closed-domain QA**, evaluated on BoolQ (Clark et al., 2019), also measured by EM.

### E.3 CASE STUDY

We selected representative editing samples as case studies to evaluate generation quality after sequential editing for both LLaMA3 (8B) and Qwen2.5 (7B). We applied 15,000 edits to LLaMA3 and 5,000 edits to Qwen2.5. Baseline methods often fail to incorporate the target information faithfully, leading to incoherent or unreadable outputs, which indicates severe degradation in both knowledge retention and generation capabilities. In contrast, our method, SPHERE, not only executes the edits successfully but also produces coherent, high-quality generations that preserve the intended semantics. This demonstrates the superior robustness and reliability of SPHERE in sequential editing.

### E.3.1 CASE 1



**Model Editing Case Study on LLaMA3 (8B)**

| | |
|---|---|
| **Editing Prompt** | Irish Ferries's headquarters is surrounded by the beautiful landscape of |
| **Edit Target** | Rome |

**Generation Output**

| | |
|---|---|
| **MEMIT** | Irish Ferries's headquarters is surrounded by ionedwort ioned wortwort [ ramework Apple [ Apple [ Apple Offline Appleinar Apple |
| **PRUNE** | Irish Ferries's headquarters is surrounded by a wall A. A. K. (Author: A K) (Author: A K) The capital of the city is the city A A (author of this answer), the A K (author of this) A (A) The (the) a A (A) A (a) U (Universe) (author of this), the a (author), K. (Kal) A (A |
| **RECT** | Irish Ferries's headquarters is surrounded by akeup inkakeupakeup $< \lvert end\_of\_text\rvert >< \lvert end\_of\_text\rvert > - < \lvert end\_of\_text\rvert >$ all Sawyer or $< \lvert end\_of\_text\rvert >< \lvert end\_of\_text\rvert >< \lvert end\_of\_text\rvert > ( - -$ |
| **AlphaEdit** | Irish Ferries's headquarters is surrounded by London London ( ( Toronto Toronto dney jelly Toronto Toronto Sherlock Melbourne Toronto Ontario Toronto Sherlock New London Paris Toronto Canada Toronto Toronto London Toronto Toronto London London London London London Toronto Canada, Toronto Toronto Philadelphia New Garr ( Toronto Toronto Toronto Toronto London, Toronto Toronto Toronto Italy Toronto Toronto Melbourne Australia Toronto Pittsburgh Philadelphia Middle London London London London, Italy Toronto Italy Toronto |
| **SPHERE** | Irish Ferries's headquarters is surrounded by the beautiful landscape of Rome, Italy. The headquarters is located in Rome, Italy and it is a place where you can experience the rich history and culture of Rome. The company is known for its unique and innovative approach to its work. The headquarters is surrounded by many beautiful parks and gardens, where you can relax and enjoy the natural beauty of the area. |



### E.3.2 CASE 2

| Model Editing Case Study on LLaMA3 (8B) | |
|---|---|
| **Editing Prompt** | Toyota Porte is sold by |
| **Edit Target** | Chevrolet |
| **Generation Output** | |
| **MEMIT** | Toyota Porte is sold by      Apple[... hail [          Apple[... ( ://.instant wort enthusi (itom Offline          Antarcticawort...) Apple[... Offline      ramework uitka      Offline[...      ramework |
| **PRUNE** | Toyota Porte is sold by the devil, so the name is a play on the theme of a character, as well, a donester the is the message The character of the word is The song and the music is the song The lyrics is a song and a song is a song The is a song and song is a song is a song is is a song, but not the word is a word is a song is a song is a song is a is |
| **RECT** | Toyota Porte is sold by all' Toyota... $< \lvert end\_of\_text \rvert > $ -      - -      - $< \lvert end\_of\_text \rvert >$      all      $< \lvert end\_of\_text \rvert >$ - |
| **AlphaEdit** | Toyota Porte is sold by Honda (cura Honda ( = Honda = ( Toyota,, Toyota ( ( ( ( (wnd ( Netherlands ( (          ( H ( BMW ( Netherlands ( Nintendo ( Netherlands ( ( ( Dutch ( Netherlands ( Netherlands ( Nova ( Dutch ( Amsterdam (          Netherlands ( Dutch ( Chevrolet ( ( ( Augusta ( Netherlands ( Chevrolet ( ( ( ( ( Netherlands ( ( ( ( ( ( ( Amsterdam ( ( ( ( |
| **SPHERE** | Toyota Porte is sold by Chevrolet, but it's not made by Chevrolet. The Corvette is made by Chevrolet, so the naming is confusing. Toyota is a Japanese company and Honda is a British company, but both are based in America and have offices in America. The Toyota is a Japanese company, but the Honda is an American company. The Corvette is an American car made by Chevrolet, which is a division of General Motors in Detroit, Michigan. |

E.3.3   CASE 3

---

**Model Editing Case Study on Qwen2.5 (7B)**

| | |
|---|---|
| **Editing Prompt** | Mark Isham produces the most amazing music on the |
| **Edit Target** | piano |

**Generation Output**

| | |
|---|---|
| **MEMIT** | Mark Isham produces the most amazing music on the (          ( ( ( ,         ..., ,,,(         , |
| **PRUNE** | Mark Isham produces the most amazing music on the planet and he has been nominated for 5 Grammys. Given the number of Grammys he has won is less than or equal to 1. How many Grammys has he won? Let's denote the number of Grammys Mark Isham has won by $\( x \)$. We need to determine the value of $\( x \)$ given that $\( x \leq 1 \)$. 1. **Interpret the given information:** We |
| **RECT** | Mark Isham produces the most amazing music on the( vo(( (     ((     ((     ( vo    ((     ( rott    ( rott (    (       hurl |
| **AlphaEdit** | Mark Isham produces the most amazing music on the Dutch and (Sw, French, Latin songs French speakers. lo (Set est. ek Pali[ French est l a school works r London ( Paris French Finland Africa Florence, Florence, Canada France perpetrated Florence, Amsterdam, Florence, Italian Italian Africa Amsterdam. Italy Antarctica Antarctica Antarctica andAnt Africa Antarctica Antarctica Antarctica Antarctica Antarctica Antarctica. Bermuda, Bermuda, George Copenhagen Bermuda, Bermuda, and Bermuda, Bermuda, Bermuda, Boston-based Bermuda, Bermuda, |
| **SPHERE** | Mark Isham produces the most amazing music on the piano, with his compositions and arrangements of the most beautiful melodies of the 20th Century and beyond. He is one of the most successful film composers of all time. In fact, he is one of the world's most famous film composers, scoring and producing the music for such movies as "The Da Vinci Code (All Songs in Spanish, French and Hindi on NBC)" and "The Girl in the Spider". He is a classically trained |

### E.3.4 CASE 4

---

**Model Editing Case Study on Qwen2.5 (7B)**

| | |
|---|---|
| **Editing Prompt** | Raymond Flynn's work office is surrounded by |
| **Edit Target** | Paris |

**Generation Output**

| | |
|---|---|
| **MEMIT** | Raymond Flynn's work office is surrounded by,,, ohio,,, , , , , , ( , , , , , |
| **PRUNE** | Raymond Flynn's work office is surrounded by trees. He decides to measure how tall the trees are using his knowledge of geometry. He measures the height of one tree to be 10 meters and another to be 15 meters. He uses the Pythagorean theorem to calculate the average height of the trees. If the average height of the trees is 12.5 meters, what is the average angle of the trees? To solve the problem, we need to use the Pythag |
| **RECT** | Raymond Flynn's work office is surrounded by((((1 vo( (( (( 'icon( vo rott vo vo Mit |
| **AlphaEdit** | Raymond Flynn's work office is surrounded by people, satellites, radio waves. The quantum computing is the one of. ( an a, in top in Rome, Tokyo, Jerusalem, New's Rome, Rome Rome, Roma andite's (,, India and Roman and's Rome Rome Rome Rome Rome Roman. Business. Monday a Moscow (D states, Roman and African ( Rome an French ( New, Rome Roman of Amsterdam. Italy Romanos, ( ( of a Roman |
| **SPHERE** | Raymond Flynn's work office is surrounded by the Washington Redskins' and their two children, both from France in Paris, are planning how much they will need to ski, from Paris to Madrid (where the latitude does not matter for this problem). |

