# OpenReview forum: "Energy-Regularized Sequential Model Editing on Hyperspheres"
_ICLR.cc/2026/Conference — ICLR 2026 Poster_

### Official Review · Reviewer_yBgS · 2025-10-26

**Soundness:** 4
**Presentation:** 3
**Contribution:** 3
**Rating:** 8
**Confidence:** 3

**Summary:**

This paper addresses catastrophic forgetting in Large Language Models (LLMs) during sequential editing. The authors propose a core hypothesis: hyperspherical uniformity of model weights is key to stability and knowledge retention. To validate it, they first use Hyperspherical Energy (HE) to quantify this uniformity, and show strong correlation between HE fluctuations and editing failures. They theoretically prove that changes in HE impose a lower bound on output perturbation, providing theoretical support for HE stability. Based on these findings, they propose SPHERE (Sparse Projection for Hyperspherical Energy-Regularized Editing). SPHERE identifies principal hyperspherical directions in pretrained weights and projects new knowledge updates into a complementary sparse subspace, aiming to inject new information without disturbing the model's core knowledge structure.

**Strengths:**

Strength 1: The main contribution of this paper lies in providing a novel geometric perspective—the disruption of "hyperspherical uniformity"—for understanding model editing failures (catastrophic forgetting), offering a plausible explanation for why models collapse.

Strength 2: The proposed method demonstrates clear advancement over existing mainstream approaches, showing strong scalability and promising potential.

**Weaknesses:**

Weakness 1: Since the method involves principal component analysis, applying it to larger models will incur significant computational overhead, yet the paper lacks relevant discussion and analysis on this issue.

Weakness 2: The paper lacks analysis and experiments regarding the settings of hyperparameters.

**Questions:**

Q1: Related to Weakness2, In particular, I noticed that different parameter choices are used for AlphaEdit and other editing methods, why?

Q2: Have the authors considered leveraging the theoretical lower bound of HE change to proactively estimate a model's potential editing capacity, or in other words, its editing limit?

---

> ### Author Response · Authors · 2025-11-20
> **Part1**
>
> We sincerely thank the reviewer for the insightful and valuable comments! We are so excited to receive many encouraging reviews recognizing **our novel geometric perspective on catastrophic forgetting through hyperspherical uniformity, as well as the SPHERE’s clear strong scalability and potential over existing approaches.** We respond to the raised questions as follows and will incorporate the recommended revisions into the updated manuscript.
>
> ---
>
> 1. **RE: “computational overhead analysis”**
>     To assess the computational overhead under the settings of our main experiment, we measured the time required for the projection step and for the entire batch-editing process. The tables below report these times along with the projection-to-total ratio for reference. It shows that the projection time accounts for only 3.31% of the whole editing time for LLaMA3 (8B), 6.71% for Qwen2.5 (7B), and 6.01% for Qwen2.5 (32B).
>
>
>     | Model       | Edit Time | Projection Time |     ratio         |
>     |-------------|-----------|------------------|------------|
>     | Qwen2.5-7B |    $535.73$       |         $35.95$        |     6.71\%         |
>     | LLaMA3-8B   |       $543.26$|           $18.00$       |         3.31\%           |
>     | Qwen2.5-32B   |       $1656.58$|           $99.60$       |         6.01\%           |
>
>
>
> 2. **RE: “sensitivity to hyperparameters cumulative ratio $\eta$ and suppression strength $\alpha$”**
>    As analyzed in Section 3 (Figure 2/3), different editing methods exhibit different magnitudes of HE fluctuation, with AlphaEdit showing a noticeably smaller increase in HE compared to the other baselines. To avoid over-suppressing its update, we therefore use a smaller suppression strength (α = 0.5) for AlphaEdit, while keeping α = 0.8 for the other methods.
>
>    We agree that assessing SPHERE’s sensitivity to the cumulative ratio η and suppression strength α is important for evaluating its robustness. To address this, we provide an expanded ablation study that systematically varies both hyperparameters over a wider range. The table below reports results for different combinations of α and η, with all other settings consistent with the main experiment. Across all configurations, our sparse projection method consistently improves the performance of its original counterpart, demonstrating that SPHERE remains robust and effective across a broad range of hyperparameters. Although the magnitude of improvement varies with the suppression strength, different editing methods or application scenarios may benefit from different (α, η) choices. Practitioners may therefore tune these hyperparameters to suit their specific baseline or use case.
>
>     | η \ α       | α = 0 |α = 0.3 | α = 0.5 | α = 0.7 |
>     |-------------|---|-------|----------|----------|
>     | η = 0    |  $10.63$  |   $10.63$    |    $10.63$     |     $10.63$     |
>     | η = 0.3     |  $10.63$ | $26.06$      |   $27.60$       |    $53.06$      |
>     | η = 0.5     | $10.63$  | $34.20$      |     $21.77$     |     $61.17$     |
>     | η = 0.7     | $10.63$ |  $22.50$      |   $29.03$       |     $89.00$     |

---

> ### Author Response · Authors · 2025-11-20
> **Part2**
>
> 3. **RE: “editing limit”**
>     We agree that estimating the “editing capacity’’ of a method is an important and exciting direction. Below we clarify what our current theory enables, and why it is hard to directly provide an editing limit.
>     While using theoretical bounds to estimate editing capacity is appealing, the mathematical formulation we derive provides only a one-sided constraint. In Theorem 1 (together with Eq. 8), we show:
>     $$
>     \Delta V \ge \frac{1}{K^2} (\Delta \text{HE})^2
>     $$
>     where $\Delta V$ represents the energy of the parameter/output perturbation. This inequality tells us: 1) Any increase in HE necessarily imposes a minimum amount of output perturbation; 2) Editing capacity is typically governed by how much perturbation $\Delta V$ the model can tolerate before degrading semantics. **However, this relationship gives us only the minimum $\Delta V$ required for a given $\Delta HE$. It does not give an upper bound on $\Delta V$, which is essential for estimating an editing limit.**
>     To estimate a model’s editing limit, one would need a formula such as:
>     $$
>     \Delta V_{min} \leq \Delta V \leq \Delta V_{max}
>     $$
>     But no such upper bound exists under current assumptions. We provided a simple counterexample shows why: consider applying a global orthogonal rotation to the weight matrix. Let $Q$ be any orthogonal matrix $(Q^TQ = I, but Q \neq I)$ and define:
>     $$
>     W_{new} = WQ, \Delta W = W(Q-I)
>     $$
>     Since orthogonal transformations preserve all pairwise angles, we have $\Delta HE = 0$, however, depending on the choice of $𝑄$, the perturbation magnitude $\Delta V$ can be arbitrarily large. **This counterexample illustrates that no upper bound on editing capacity can be inferred from HE alone, and therefore an editing limit cannot be derived from our current bound.**
>     Although impossible using HE alone, we believe more granular structural measures could potentially estimate editing capacity in the future.

---

> > ### Comment · Reviewer_yBgS · 2025-11-25
> >
> > Thank you for your response and the additional experiments. The PCA component accounts for a small fraction of the editing time, and this proportion remains consistent across models of different scales within the same family. The hyperparameter experiments and the restatement of the HE theory are also clear. I will maintain my positive score.

---

> > > ### Author Response · Authors · 2025-11-25
> > >
> > > Thank you for your response! We are committed to further incorporate these clarifications into the next revision.

---

### Official Review · Reviewer_89en · 2025-10-29

**Soundness:** 3
**Presentation:** 3
**Contribution:** 3
**Rating:** 6
**Confidence:** 3

**Summary:**

### Summary

This paper addresses performance degradation in sequential model editing, where successive updates to a large language model often lead to catastrophic forgetting. The authors hypothesize that this failure is linked to the disruption of "hyperspherical uniformity", the geometric distribution of neuron weights, which they quantify using Hyperspherical Energy.
They provide empirical evidence of a strong correlation between HE fluctuations and editing failures and offer a theoretical proof that HE changes establish a lower bound on knowledge degradation. Based on this, they propose SPHERE, a regularization strategy that preserves stability by identifying the principal directions of pretrained weights and projecting new knowledge updates onto a complementary sparse space. Experiments demonstrate that SPHERE significantly improves editing capability over strong baselines while better preserving the models' general abilities.



### Advantages

* The paper provides a novel and intuitive analysis connecting the geometric concept of hyperspherical uniformity with the practical problem of catastrophic forgetting in sequential editing.
* The authors support their empirical observations with a theoretical proof establishing that changes in Hyperspherical Energy create a lower bound for the degradation of pretrained knowledge.
* The proposed SPHERE method not only outperforms existing baselines in large-scale sequential editing but also functions as a plug-and-play enhancement that significantly boosts the performance of other editing techniques.




### Drawbacks and Questions

* The proposed method relies on estimating the principal hyperspherical directions of the weight matrices, which involves an eigendecomposition that may introduce computational overhead not fully analyzed in the paper.
In this case, would it be possible to conduct an experiment analyzing the computational cost (e.g., latency or FLOPS) of applying the SPHERE projection compared to the base editing methods, particularly as the model size or the number of edits increases?



* The effectiveness of the sparse space projection appears dependent on two key hyperparameters, the cumulative ratio $\eta$ and the suppression strength $\alpha$, but the paper provides limited ablation studies on how sensitive the model's performance is to different values for these parameters.
Thus, could the authors provide an additional ablation study showing how editing performance (Efficacy and Specificity) varies across a wider range of $\eta$ and $\alpha$ values to better understand this sensitivity?



* The theoretical analysis linking HE changes to output perturbation relies on simplifying assumptions, such as "orthonormal inputs" and "small perturbations," which may not fully capture the dynamics of real-world editing scenarios.
Thus, could an empirical experiment be designed to measure the orthonormality of the actual input key vectors ($K_0$) or the magnitude of perturbations ($\Delta W$) in practice, to assess how well these assumptions hold during sequential editing?

**Strengths:**

Please see above

**Weaknesses:**

Please see above

**Questions:**

Please see above

---

> ### Author Response · Authors · 2025-11-20
> **Part1**
>
> We sincerely thank the reviewer for the insightful and valuable comments! We are so excited to receive many encouraging reviews recognizing **our novel and intuitive analysis linking hyperspherical uniformity to catastrophic forgetting, the strong theoretical support establishing bounds on knowledge degradation, and the SPHERE's superior performance and plug-and-play utility in large-scale sequential editing.** We respond to the raised questions as follows and will incorporate the recommended revisions into the updated manuscript.
>
> ---
>
> 1. **RE: "cost with model size and edit size increase"**
>     The principal hyperspherical directions are recomputed for each batch edit rather than incurred as a one-time cost. To assess the computational overhead under the settings of our main experiment, we measured the time required for the projection step and for the entire batch-editing process. The tables below report these times along with the projection-to-total ratio for reference. It shows that the projection time accounts for only 3.31% of the whole editing time for LLaMA3 (8B), 6.71% for Qwen2.5 (7B), and 6.01% for Qwen2.5 (32B). Moreover, the computational cost of SPHERE remains stable as the editing size increases, making the overhead highly acceptable in practical batch-editing scenarios.
>
>     | Model       | Edit Time | Projection Time |     ratio         |
>     |-------------|-----------|------------------|------------|
>     | Qwen2.5-7B |    $535.73$       |         $35.95$        |     6.71\%         |
>     | LLaMA3-8B   |       $543.26$|           $18.00$       |         3.31\%           |
>     | Qwen2.5-32B   |       $1656.58$|           $99.60$       |         6.01\%           |
>
>
>     | Model       | 0-500 edits | 500-1k | 1k-1.5k | 1.5k-2k | 2k-3k | Avg |
>     |-------------|--------------|-----------|-----------|-----------|-------------|-----|
>     | Qwen2.5-7B  |       $6.92\%$       |    $8.89\%$     |     $8.35\%$     |    $7.02\%$     |      $5.35\%$       |  $6.71\%$|
>     | LLaMA3-8B   |      $2.13\%$    |       $2.35\%$  |      $3.54\%$   |   $5.81\%$        |     $4.03\%$        |   $3.31\%$  |
>
>
>
>
> 2. **RE: “sensitivity to hyperparameters cumulative ratio $\eta$ and suppression strength $\alpha$”**
>
>    As analyzed in Section 3 (Figure 2/3), different editing methods exhibit different magnitudes of HE fluctuation, with AlphaEdit showing a noticeably smaller increase in HE compared to the other baselines. To avoid over-suppressing its update, we therefore use a smaller suppression strength (α = 0.5) for AlphaEdit, while keeping α = 0.8 for the other methods.
>
>    We agree that assessing SPHERE’s sensitivity to the cumulative ratio η and suppression strength α is important for evaluating its robustness. To address this, we provide an expanded ablation study that systematically varies both hyperparameters over a wider range. The table below reports results for different combinations of α and η, with all other settings consistent with the main experiment. Across all configurations, our sparse projection method consistently improves the performance of its original counterpart, demonstrating that SPHERE remains robust and effective across a broad range of hyperparameters. Although the magnitude of improvement varies with the suppression strength, different editing methods or application scenarios may benefit from different (α, η) choices. Practitioners may therefore tune these hyperparameters to suit their specific baseline or use case.
>
>     | η \ α       | α = 0 |α = 0.3 | α = 0.5 | α = 0.7 |
>     |-------------|---|-------|----------|----------|
>     | η = 0    |  $10.63$  |   $10.63$    |    $10.63$      |     $10.63$     |
>     | η = 0.3     |  $10.63$ | $26.06$      |   $27.60$       |    $53.06$      |
>     | η = 0.5     | $10.63$  | $34.20$      |     $21.77$     |     $61.17$     |
>     | η = 0.7     | $10.63$ |  $22.50$      |   $29.03$       |     $89.00$     |

---

> > ### Author Response · Authors · 2025-11-20
> > **Part2**
> >
> > 3. **RE: "validation of the assumptions in the theoretical analysis"**
> >
> >     We acknowledge that our theoretical derivation relies on simplifying assumptions to establish the inequality relationship between Hyperspherical Energy change ($\Delta \text{HE}$) and output perturbation ($\Delta V$). Below, we clarify why these assumptions are mathematically reasonable and empirically valid in practical editing scenarios.
> >
> >     **(1) Justification for the "Orthonormal Inputs" Simplification**
> >
> >     In our derivation, we assume the input key vectors $\{k_i\}$ form an orthonormal basis.
> >     - **Theoretical Abstraction:** This is a standard reduction in theoretical analysis to isolate the intrinsic geometry of the weight space from the data covariance structure. By assuming orthonormality, we can simplify the energy of the output perturbation directly to the Frobenius norm of the weight update: $\Delta V = \|\Delta W\|_F^2$.
> >     - **Qualitative Validity:** While real-world keys $\{k_i\}$ may not be strictly orthonormal, they typically span a subspace where the perturbation magnitude is linearly related to the weight change. This assumption allows us to derive a **general lower bound** that characterizes the stability of the weight geometry itself. Relaxing this assumption would introduce a scaling factor dependent on the condition number of the key matrix $K$, but it would not fundamentally alter the correlation between $\Delta \text{HE}$ and model stability.
> >
> >     **(2) Validation of the "Small Perturbations" Assumption**
> >
> >     We assume the norm of the perturbation $\Delta w_i$ is sufficiently small to justify the use of the first-order Taylor Expansion (Eq. 37, 38) while ignoring higher-order terms.
> >     - **Definition of Model Editing:** The fundamental goal of model editing is to inject specific knowledge with minimal impact on the overall model. Unlike pre-training, successful editing algorithms are explicitly regularized to produce minimal parameter updates ($\Delta W$). If $\|\Delta W\|$ were large, the process would constitute catastrophic forgetting or re-initialization rather than true "editing", contradicting the objective of “editing” and making HE preservation irrelevant.
> >     - **Empirical Verification:** To validate this, we measured the relative magnitude of the perturbation $\|\Delta W\|_F / \|W\|_F$ and the relative error of the Taylor approximation in our experiments. As shown in the table, the relative perturbation magnitude remains below **11%** both for row-wise and overall (matrix-level) perturbations. Since the function is continuous, the impact of the ignored second-order (and higher-order) terms is negligible (empirically < 1%). This confirms that the "small perturbation" assumption holds firmly in sequential editing scenarios.
> >
> >         |      | Ratio |
> >         |-------------|---|
> >         | Row-wise    |  7.35\%  |
> >         | Overall    |  7.37\%  |
> >
> >         | Layer      | Row Ratio | Overall Ratio |
> >         |-------------|---|-------|
> >         | 4    |  5.93\%  |   5.89\%     |
> >         | 5     |  5.80\%  | 5.79\%       |
> >         | 6     | 6.47\%   | 6.45\%       |
> >         | 7     | 7.74\%  |  7.72\%       |
> >         | 8     | 10.91\%  |  10.89\%       |

---

> > > ### Comment · Reviewer_89en · 2025-11-21
> > >
> > > I thank the authors for including additional empirical results to address my questions. I have updated my scores accordingly.

---

> > > > ### Author Response · Authors · 2025-11-21
> > > >
> > > > We sincerely thank you for taking the time to reevaluate our work and for the positive update to your score!

---

### Official Review · Reviewer_Xwiv · 2025-10-30

**Soundness:** 3
**Presentation:** 4
**Contribution:** 2
**Rating:** 4
**Confidence:** 4

**Summary:**

The paper focuses on catastrophic forgetting in sequential knowledge editing for LLMs. The authors argue that maintaining the hyperspherical uniformity of the model’s internal weights is key to balancing the preservation of pretrained knowledge with the integration of new edits. They employ hyperspherical energy to quantify weight uniformity and use this metric to develop HE-driven regularization strategies that stabilize the editing process and mitigate forgetting.

**Strengths:**

- The paper provides both empirical and theoretical analyses showing how the hyperspherical uniformity of an LLM’s internal weights affects sequential knowledge editing—a novel and valuable perspective for this problem.
- The paper introduces SPHERE, an HE-driven regularization strategy that stabilizes neuron weight distributions, thereby preserving prior knowledge while enabling reliable sequential updates.

**Weaknesses:**

- The AlphaEdit baseline appears much lower than its reported results; please reconcile this via implementation details, protocol alignment, multi-seed statistics, and a brief reproduction-gap analysis (ideally with official code).
- Please clearly articulate how SPHERE differs from AlphaEdit conceptually, algorithmically.

**Questions:**

Please see weaknesses

---

> ### Author Response · Authors · 2025-11-20
>
> We are so excited to receive many encouraging reviews recognizing **our novel empirical and theoretical perspective linking hyperspherical uniformity to sequential editing stability, as well as SPHERE’s effectiveness as an HE-driven regularization method that preserves prior knowledge while enabling reliable sequential updates.** We respond to the raised questions as follows and will incorporate the recommended revisions into the updated manuscript.
>
> ---
>
> 1. **RE: “result difference from AlphaEdit”**
>
>    All baseline experiments, including AlphaEdit, were conducted using their official codebases and default hyperparameters. Our evaluation is substantially more challenging, as we **perform far more sequential edits**. In addition, we adopted a **more rigorous and consistent metric across datasets**, which led to differences from the originally reported AlphaEdit results. Specifically:
>
>    - **Different numbers of sequential edits.** Our evaluation performs 15,000 sequential edits on LLaMA3-8B, whereas the AlphaEdit paper reports results with only 2,000 edits. We chose 15,000 edits because real-world model maintenance often requires large-scale updates, where degradation typically accumulates. Evaluating on substantially more edits therefore provides a more rigorous and realistic assessment of the stability and scalability of a method.
>    - **Evaluation metric.** We use a more rigorous and consistent metric: average top-1 accuracy across all datasets (reported in Appendix D). In contrast, AlphaEdit used success rate for CounterFact, which only measures whether the edited answer is preferred over the original, a much weaker criterion, while using top-1 accuracy for ZsRE. Top-1 accuracy directly measures correctness and therefore provides a more robust assessment of editing quality.
>
> 2. **RE: “conceptual difference from AlphaEdit”**
>
>    Although both AlphaEdit and SPHERE use projections into a subspace, the nature and purpose of these nullspaces are fundamentally different.
>
>    - **AlphaEdit** constructs **a semantic nullspace** tied to the original knowledge key set $K_0$. It projects the update weight ∆W into the nullspace of the original knowledge key set $K_0$, minimizing the output perturbation $∆V = ∆W K_0$.
>    - In contrast, **SPHERE** constructs **a geometric subspace** tied to the weight matrix’s structure. It places no constraint relative to the original knowledge key set. Instead, it projects the update weight $∆W$ into a hyperspherical subspace that is orthogonal to the weight’s principal hyperspherical directions, preserving the angular geometry of the weight matrix and preventing the loss of weight diversity. This geometric subspace is structure-specific and serves to maintain geometric integrity rather than semantic behavior.
>
>    Conceptually, AlphaEdit preserves the model’s semantic outputs but can introduce substantial geometric distortion. Intuitively, it warps the weight space so that the weights are no longer uniformly distributed, disrupting the geometric structure that supports good generalization. In contrast, SPHERE maintains the geometric integrity of the weight space. The two methods are complementary because SPHERE counteracts the geometric collapse that AlphaEdit alone allows, leading to significantly improved generalization retention as demonstrated by our results in Section 5.
>
> 3. **RE: “algorithmical difference from AlphaEdit”**
>
>    We elaborate the algorithmical difference between AlphaEdit and SPHERE as follows.
>
>    **AlphaEdit** aims to minimize the output perturbation $∆V = ∆WK_0$ by projecting $∆W$ into the nullspace of $K_0$, thereby $∆WK_0 = 0$. The following is the general algorithmical flow:
>    - **Extract the original knowledge key vector set $K_0$.** Identify the original knowledge key vector set $K_0$ that approximates the model’s stored knowledge.
>    - **Construct the nullspace projection.** Compute a projection matrix that forces any update ∆W to lie in the nullspace of $K_0$.
>    - **Edit within the nullspace.** Apply the edit inside this constrained space, along with keeping updates bounded and preserving the newly inserted knowledge.
>
>    In contrast, **SPHERE** projects ∆W into a sparse space complementary to the principal hyperspherical directions (*U* in Eq. 11) of the pretrained weight matrices, preventing the collapse of hyperspherical uniformity. The following is the general algorithmical flow:
>     - **Estimate principal hyperspherical direction.** From the edited weight matrix W, SPHERE computes its geometric principal directions (Eq. 10).
>     - **Construct the sparse space projection.** Based on the geometric principal directions, SPHERE extracts its sparse hyperspherical subspace and constructs the corresponding projection matrix $P$ (Eq. 12).
>     - **Edit within the sparse space.** Given a perturbation matrix $∆W$ produced by any editing method, it is projected onto the sparse space, and then combined it with the original weight matrix.

---

> ### Author Response · Authors · 2025-11-26
>
> Dear Reviewer,
>
> We hope everything is going well for you. We are writing to gently follow up on our rebuttal and would be glad to provide any additional clarification if needed.
> Thank you very much for your time and effort.
>
> Authors

---

### Official Review · Reviewer_U4hX · 2025-10-30

**Soundness:** 3
**Presentation:** 4
**Contribution:** 3
**Rating:** 8
**Confidence:** 3

**Summary:**

This paper tackles the problem of performance degradation in sequential model editing. The authors propose that editing failures like catastrophic forgetting are linked to the disruption of the hyperspherical uniformity of neuron weights. They use Hyperspherical Energy or HE to quantify this uniformity and provide both empirical and theoretical evidence linking HE instability to knowledge degradation. Based on this they introduce SPHERE a regularization strategy. SPHERE works by identifying the principal directions of the pretrained weight matrix and then projecting the edit updates onto the orthogonal complement of this space. This approach aims to integrate new knowledge while minimizing disruption to the models original weight geometry. Experiments on LLaMA3-8B and Qwen2.5-7B show that SPHERE outperforms existing methods in long-sequence editing particularly in preserving the models general abilities.

**Strengths:**

1. The core idea of linking sequential editing stability to hyperspherical uniformity is a novel and insightful contribution. It provides a new and intuitive explanation for why models collapse during repeated edits.

2. The paper does a very good job of supporting this central hypothesis. Its not just an idea it's backed by strong empirical correlations and a formal theoretical analysis that bounds knowledge degradation by HE dynamics.

3. The SPHERE method itself is elegant. The strategy of projecting updates into a sparse space complementary to the principal weight directions is a clean and direct implementation of the papers main hypothesis.

4. The experimental results are impressive. SPHERE shows a clear advantage in large-scale sequential editing and very importantly in preserving the models general abilities which is a common failure point for other methods. The fact that it also works as a plug-and-play enhancement is a significant practical strength.

**Weaknesses:**

1. The link between the HE motivation and the method feels loose. HE is defined by pairwise angular relationships but the method identifies its space using a variance approach in Eq 10 that looks just like PCA. It's not obvious why high-variance directions are the most important ones for preserving HE.

2. The paper needs to better distinguish its novelty from AlphaEdit. AlphaEdit also uses null-space projection. The conceptual advantage of projecting away from weight directions versus previous knowledge directions isn't fully explored.

3. I'm concerned about the SVD scalability. The paper doesn't discuss the cost. Is it a one-time computation? If so does that original principal space remain the correct one to avoid after thousands of edits?

4. The method introduces $\eta$ and $\alpha$ but lacks a sensitivity analysis. The values seem set arbitrarily for example $\alpha$ is 0.5 for AlphaEdit but 0.8 for others. This makes it hard to judge robustness.

**Questions:**

1. Can you elaborate on the SVD cost? Is it a one-time cost? I'm wondering if the original models principal directions are still the most important ones to preserve after 10000 edits.

2. Why is preserving the principal directions of the weight matrix the right way to stabilize HE? I'm trying to square this variance-based method with the angular-distance-based HE metric.

3. How did you select the $\alpha$ and $\eta$ values? The performance seems like it would be sensitive to these. The choice of $\alpha$ is different for different baselines when you use SPHERE as a plug-in. Could you provide an ablation on this?

---

> ### Author Response · Authors · 2025-11-20
> **Part1**
>
> We sincerely thank the reviewer for the insightful and valuable comments! We are so excited to receive many encouraging reviews recognizing that our paper **makes a novel contribution by linking sequential editing stability to hyperspherical uniformity, is supported by strong empirical and theoretical analysis, introduces the elegant and principled SPHERE method, and presents impressive experimental results.** We respond to the raised questions as follows and will incorporate the recommended revisions into the updated manuscript.
>
> ---
>
> 1. **RE: “the link between HE motivation and the method” and “why is preserving the principal directions of the weight matrix the right way to stabilize HE”**
>
>    Hyperspherical energy (HE) is defined as the sum of pairwise angular terms between weight vectors, and any perturbation of the weights alters these angles to varying degrees. To formalize this, for each perturbation direction in weight space ($\hat{v}$ in Eq. 10), we measure how strongly it affects the angle between every pair of neurons. These pairwise angular sensitivities are then aggregated to construct a covariance-like matrix that reflects how much each direction influences the overall HE (Eq. 10). Directions with high “variance” correspond to perturbations that cause large changes in HE, while directions with low “variance” have minimal impact. Preserving the high-variance directions is therefore essential, and our method explicitly accounts for this. We will clarify this construction and its interpretation more fully in the revision.
>
>
>
> 2. **RE: “comparison between SPHERE and AlphaEdit”**
>
>    We elaborate their differences from the conceptual and algorithmical perspectives respectively as follows.
>
>    (1) **Conceptual difference**: Although both AlphaEdit and SPHERE use projections into a subspace, the nature and purpose of these nullspaces are fundamentally different.
>
>    - **AlphaEdit** constructs **a semantic nullspace** tied to the original knowledge key set $K_0$. It projects the update weight ∆W into the nullspace of the original knowledge key set $K_0$, minimizing the output perturbation $∆V = ∆W K_0$.
>    - In contrast, **SPHERE** constructs **a geometric subspace** tied to the weight matrix’s structure. It places no constraint relative to the original knowledge key set. Instead, it projects the update weight $∆W$ into a hyperspherical subspace that is orthogonal to the weight’s principal hyperspherical directions, preserving the angular geometry of the weight matrix and preventing the loss of weight diversity. This geometric subspace is structure-specific and serves to maintain geometric integrity rather than semantic behavior.
>
>    Conceptually, AlphaEdit preserves the model’s semantic outputs but can introduce substantial geometric distortion. Intuitively, it warps the weight space so that the weights are no longer uniformly distributed, disrupting the geometric structure that supports good generalization. In contrast, SPHERE maintains the geometric integrity of the weight space. The two methods are complementary because SPHERE counteracts the geometric collapse that AlphaEdit alone allows, leading to significantly improved generalization retention as demonstrated by our results in Section 5.
>
>    (2) **Algorithmical difference**:
>
>    **AlphaEdit** aims to minimize the output perturbation $∆V = ∆WK_0$ by projecting $∆W$ into the nullspace of $K_0$, thereby $∆WK_0 = 0$. The following is the general algorithmical flow:
>    - **Extract the original knowledge key vector set $K_0$.** Identify the original knowledge key vector set $K_0$ that approximates the model’s stored knowledge.
>    - **Construct the nullspace projection.** Compute a projection matrix that forces any update ∆W to lie in the nullspace of $K_0$.
>    - **Edit within the nullspace.** Apply the edit inside this constrained space, along with keeping updates bounded and preserving the newly inserted knowledge.
>
>    In contrast, **SPHERE** projects ∆W into a sparse space complementary to the principal hyperspherical directions (*U* in Eq. 11) of the pretrained weight matrices, preventing the collapse of hyperspherical uniformity. The following is the general algorithmical flow:
>     - **Estimate principal hyperspherical direction.** From the edited weight matrix W, SPHERE computes its geometric principal directions (Eq. 10).
>     - **Construct the sparse space projection.** Based on the geometric principal directions, SPHERE extracts its sparse hyperspherical subspace and constructs the corresponding projection matrix $P$ (Eq. 12).
>     - **Edit within the sparse space.** Given a perturbation matrix $∆W$ produced by any editing method, it is projected onto the sparse space, and then combined it with the original weight matrix.

---

> ### Author Response · Authors · 2025-11-20
> **Part2**
>
> 3. **RE: “SVD scalability”**
>
>    The principal hyperspherical directions are recomputed for each batch edit rather than incurred as a one-time cost. To assess the computational overhead under the settings of our main experiment, we measured the time required for the projection step and for the entire batch-editing process. The tables below report these times along with the projection-to-total ratio for reference. It shows that the projection time accounts for only 3.31% of the whole editing time for LLaMA3 (8B), 6.71% for Qwen2.5 (7B), and 6.01% for Qwen2.5 (32B).
>
>
>     | Model       | Edit Time | Projection Time |     ratio         |
>     |-------------|-----------|------------------|------------|
>     | Qwen2.5-7B |    $535.73$       |         $35.95$        |     6.71\%         |
>     | LLaMA3-8B   |       $543.26$|           $18.00$       |         3.31\%           |
>     | Qwen2.5-32B   |       $1656.58$|           $99.60$       |         6.01\%           |
>
>
>     In addition, we examined whether the original principal directions remain effective after a large number of sequential edits. We compared top-1 accuracy under two settings: (1) our current method to recalculate the projection matrix at every batch-editing step, and (2) reusing the initial projection matrix without further updates. The substantial performance gap between these settings shows that continuously updating the projection matrix yields markedly better editing performance. This also confirms that sequential edits significantly shift the principal directions of the weights over time, underscoring the necessity of maintaining spherical uniformity in our method throughout the editing process.
>
>     | Model       | Stabe SPHERE | SPHERE |
>     |-------------|-------|--------|
>     | LLaMA3-8B   |    $2.23$   |   $57.65$     |
>
>
>
>
> 4. **RE: “ablation analysis on the hyperparameters of SPHERE”**
>
>    As analyzed in Section 3 (Figure 2/3), different editing methods exhibit different magnitudes of HE fluctuation, with AlphaEdit showing a noticeably smaller increase in HE compared to the other baselines. To avoid over-suppressing its update, we therefore use a smaller suppression strength (α = 0.5) for AlphaEdit, while keeping α = 0.8 for the other methods.
>
>    We agree that assessing SPHERE’s sensitivity to the cumulative ratio η and suppression strength α is important for evaluating its robustness. To address this, we provide an expanded ablation study that systematically varies both hyperparameters over a wider range. The table below reports results for different combinations of α and η, with all other settings consistent with the main experiment. Across all configurations, our sparse projection method consistently improves the performance of its original counterpart, demonstrating that SPHERE remains robust and effective across a broad range of hyperparameters. Although the magnitude of improvement varies with the suppression strength, different editing methods or application scenarios may benefit from different (α, η) choices. Practitioners may therefore tune these hyperparameters to suit their specific baseline or use case.
>
>     | η \ α       | α = 0 |α = 0.3 | α = 0.5 | α = 0.7 |
>     |-------------|---|-------|----------|----------|
>     | η = 0    |  $10.63$  |   $10.63$    |    $10.63$     |     $10.63$     |
>     | η = 0.3     |  $10.63$ | $26.06$      |   $27.60$       |    $53.06$      |
>     | η = 0.5     | $10.63$  | $34.20$      |     $21.77$     |     $61.17$     |
>     | η = 0.7     | $10.63$ |  $22.50$      |   $29.03$       |     $89.00$     |

---

### Author Response · Authors · 2025-12-01
**A Summary of Review, Rebuttal, and Discussion**

Dear Area Chair,

We are deeply regretful about the recent information leakage within the community. To support a fair and thorough evaluation, we would like to summarize the reviews and our rebuttal, and draw attention to a key mismatch identified in one of the reviews.

Overall, reviewers unanimously recognized the paper’s core contribution for **novel geometric perspective linking hyperspherical uniformity to sequential editing stability and catastrophic forgetting** (U4hX, Xwiv, 89en, yBgS), supported by **strong theoretical analysis** (U4hX, Xwiv, 89en) and **comprehensive empirical validation** (U4hX, yBgS, 89en). They highlighted SPHERE’s **elegant, principled, and intuitive design** (U4hX, 89en) and noted its effectiveness in preserving prior knowledge while enabling reliable sequential updates (Xwiv). Reviewers further praised SPHERE’s **plug-and-play utility** and superior performance in large-scale sequential editing (U4hX, yBgS, 89en).

During rebuttal, we have carefully addressed the main concerns raised by the reviewers, including (1) the computational cost of our approach, (2) additional ablations on key hyperparameters, and (3) the conceptual and algorithmic distinctions between SPHERE and AlphaEdit. Our extended analysis was positively received by reviewers 89en and yBgS.

  - **Reviewer 89en** recognized the added empirical results presented during the rebuttal and subsequently raised their score **from 6 to 8 on 20 Nov. 2025** at 20:14 ETS, as documented in their comment. Given that this update took place approximately one week prior to the leak incident, we believe this update is trustworthy and reflects the strengths of our work.
  - **Reviewer yBgS** acknowledged that our theoretical and experimental clarifications fully addressed their concerns. Accordingly, he **maintained his positive score of 8** while increasing confidence score **from 3 to 4 on 25 Nov. 2025** at 02:54 EST, as recorded in updated comment. Since this adjustment was made before the leak incident, we believe it remains reliable.
  - **Reviewer Xwiv** gave a rating of 4, although their comments focused solely on asking about “the difference between SPHERE and the baseline AlphaEdit,” without engaging with our method’s core ideas, experiments, or contributions. This indicates a clear **review mismatch**, as the assessment does not correspond to the actual substance of the paper. Nevertheless, we have thoroughly clarified the empirical, conceptual, and algorithmic differences in our rebuttal. Unfortunately, Reviewer Xwiv has **remained silent** and has not responded to these clarifications. We kindly ask the AC to take this context into account during the review process.

Thank you again for your time and for your service to the community. Please let us know if any additional clarification is needed.

Best regards,

Authors

---

### Meta-Review · Area_Chair_Js7n · 2026-01-07

**Summary:**

The reviewers consistently recognized the paper’s methodological contribution, noting that the proposed SPHERE method is novel and well-supported by both theoretical analysis and empirical validation.

While several concerns were raised during the review and rebuttal phases, these have been addressed in the rebuttal, particularly the key issue regarding the distinctions between SPHERE and the AlphaEdit method.

**Reviewer Concerns:**

The rebuttal addressed the conceptual and algorithmic distinctions between the proposed method, SPHERE, and prior work, AlphaEdit, as well as the computational cost of SPHERE and additional ablation studies on key hyperparameters.

No significant issues remain outstanding.

**Reviewer Scores:**

- Reviewer U4hX explicitly stated that they will maintain their rating of 8.
- Reviewer Xwiv is likely to increase their rating to 6.
- Reviewer 89en raised their rating from 6 to 8 prior to the incident.
- Reviewer yBgS will retain their rating of 8.

---

### Decision · Program_Chairs · 2026-01-26

Accept (Poster)